# Distributionally Robust Set Representation Learning Under Inference-Time Element Corruption

Yankai Chen [1 2]   Hanrong Zhang [3]   Bowei He [1 2 *]   Philip S. Yu [3]   Xue Liu [1 2]

## Abstract

Standard Set Representation Learning methods typically excel on curated data but often overlook the challenge of *Inference-time Element Corruption*. This refers to scenarios where deployed models encounter element-level degradations, such as outliers or missing components, that may distort set representation and degrade performance. We propose SW-DRSO, a distributionally robust optimization framework tailored for sets. Rather than minimizing loss solely on observed training data, SW-DRSO optimizes a tractable surrogate of the worst-case expected loss over a family of plausible inference-time variations. We introduce a barycentric adversary that approximates the intractable search over corrupted sets by a differentiable training-time optimization over simplex weights. Extensive experiments across four tasks demonstrate that SW-DRSO effectively enhances robustness against corruption while maintaining high overall performance.

## 1. Introduction

Set Representation Learning (SRL) aims to encode a unordered collection of elements into a vector while preserving essential set-theoretic properties (Zaheer et al., 2017; Skianis et al., 2020). This capability is crucial across diverse domains, from processing point clouds (Qi et al., 2017) and modeling protein structures (NaderiAlizadeh & Singh, 2025), to enabling batch retrieval and recommendation (Li et al., 2021; Zhang et al., 2022; Qiu et al., 2024; Chen et al., 2023b; He & Ma, 2024). Recent methods employ attention mechanisms (Lee et al., 2019), optimal transport formulations (Skianis et al., 2020; Mialon et al., 2021), and

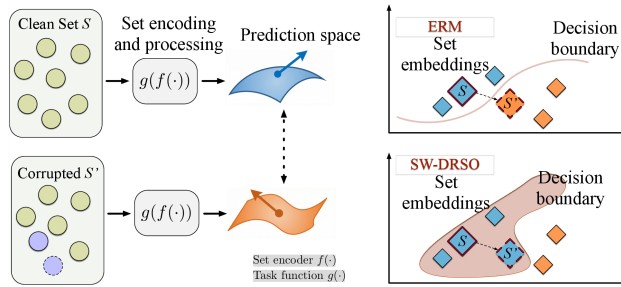

**(a)** Inference-time Element Corruption   **(b)** ERM vs SW-DRSO

*Figure 1.* Illustration of inference-time element corruption. (a) The corruption leads to variations in the prediction space. (b) While ERM performance degrades on the corrupted set $S'$, our SW-DRSO enhances robustness by optimizing the decision boundary.

sophisticated aggregation strategies (Zhang et al., 2020) to capture complex set structures while adhering to preserve permutation invariance and handle variable cardinality.

While standard SRL methods excel on *curated* data, they often overlook a robustness challenge in practical environments: *Inference-time element corruption*. Specifically, it refers to scenarios where models trained on clean and complete sets, but are deployed on inputs with element-level degradation at inference time. Such corruption is typically sparse and localized: it affects only a small fraction of elements without altering the overall set theme, i.e., *label-preserving*. Common examples include outlier points that appear in point clouds due to sensor errors (Rusu et al., 2008). Despite their sparsity, as shown in Figure 1(a), even a few corrupted elements may impact the set encoder and task processing to produce unreliable predictions. For instance, attention-based pooling is vulnerable to outliers that attract disproportionate weights, thereby dominating the global representation (Zhou et al., 2022).

To handle such inference-time uncertainty, *Distributionally Robust Optimization (DRO)* offers a principled framewor (Sagawa et al., 2019; Bertsimas et al., 2019). By minimizing the worst-case expected loss over a family of plausible distributions, DRO can hedge against unknown inference-tiem corruption patterns (Rahimian & Mehrotra,

*Corresponding author. [1]MBZUAI [2]McGill University [3]University of Illinois Chicago. Correspondence to: Yankai Chen <yankaichen@acm.org>, Bowei He <Bowei.He@mbzuai.ac.ae>.

*Proceedings of the 43[rd] International Conference on Machine Learning*, Seoul, South Korea. PMLR 306, 2026. Copyright 2026 by the author(s).

2019). This approach offers a natural extension beyond standard Empirical Risk Minimization (ERM), which optimizes performance only on nominal training distributions and may fail to generalize when inference-time inputs deviate from the training regime.

However, applying DRO directly to SRL problems faces fundamental computational challenges. First, defining the ambiguity regions[1] is non-trivial for set-structured data. Unlike continuous inputs where ambiguity regions are naturally constructed via norm-bounded perturbations (Rahimian & Mehrotra, 2022), sets are discrete and combinatorial. Second, even with a well-defined ambiguity region, solving the inner maximization in conventional DRO formulation to identify the worst cases is computationally prohibitive. This typically requires solving discrete combinatorial optimization with factorial complexity (Gao et al., 2024; Kuhn et al., 2019), making the training process unscalable.

To address these challenges, we propose **SW-DRSO** (Sliced-Wasserstein Distributionally Robust Set Optimization), a scalable DRO framework designed for SRL. As shown in Figure 1(b), it improves the robustness with better optimized decision boundary compared with conventional ERM. Specifically, we first represent sets as empirical measures (Muandet et al., 2012) and define the corruption region via the Sliced-Wasserstein (SW) metric (Bonneel et al., 2015; Naderializadeh et al., 2021; Amir & Dym, 2025). This formulation enables flexible modeling of corrupted sets without requiring discrete enumeration, while the geometric properties of the SW metric facilitate set encoding and ambiguity region construction. Second, we introduce a data synthesis strategy that approximates the worst-case optimization through Wasserstein barycentric adversaries (Agueh & Carlier, 2011). By synthesizing barycenters from local neighborhoods, we transform the intractable search over the combinatorial set space into efficient optimization over a *low-dimensional probability simplex* (i.e., the convex hull of mixing weights). Our analysis shows that this differentiable parameterization approximates the inner supremum better than discrete optimization. To empirically validate SW-DRSO, we extensively evaluate SW-DRSO across four diverse downstream tasks spanning varied data modalities and distinct corruption patterns, comparing it against state-of-the-art set representation baselines. In summary, our main contributions are three-fold:

- We propose SW-DRSO to enable effective distributionally robust optimization for Set Representation Learning.

- We formulate and optimize a tractable ambiguity region through barycentric data synthesis, which converts the

___
[1]The family of plausible corrupted distributions. We use "region" instead of the standard term "ambiguity set" to avoid confusion with the set-structured data itself.

computationally prohibitive worst-case search into an efficient, differentiable optimization task.

- Our results show that SW-DRSO consistently outperforms baselines, enhancing robustness against severe corruptions without compromising accuracy on clean data.

## 2. Related Work

**Set Representation Learning (SRL).** Representation learning aims to transform raw structured objects into compact embeddings that preserve task-relevant semantics for downstream prediction, retrieval, clustering, and decision making (Zhang et al., 2022; Chen et al., 2023b; 2024; Li et al., 2026a;c;b). Early methods focus on mapping sets to a Hilbert space using permutation-invariant pooling operations (Zaheer et al., 2017; Skianis et al., 2020; Murphy et al., 2018; Lee et al., 2019; Zhang et al., 2020; 2019). Some architectures learn optimal permutations directly (Zhang et al., 2019; Rezatofighi et al., 2018) or establish canonical ordering through sorting (Zhang et al., 2020). Attention-based mechanisms have also become prominent; Lee et al. (2019) adapted the Transformer architecture (Vaswani et al., 2017) for set data, while Jaegle et al. (2021) utilized cross-attention for permutation invariance. Alternative strategies include minimizing discrepancies between sets (Mialon et al., 2021), box embeddings (Lee et al., 2022), and fuzzy representations (Xu et al., 2025). For geometric alignments, Skianis et al. (2020) pioneered Optimal Transport via Bipartite Matching, while Naderializadeh et al. (2021); Chen et al. (2026b) introduced Sliced-Wasserstein embedding to capture distributional topology. Recent advances include meta-learning (Guo et al., 2021; Lee et al., 2023), injective embeddings like FSW (Amir & Dym, 2025), quantum modeling (Vargas-Calderón, 2025), and protein language applications (NaderiAlizadeh & Singh, 2025). We adopt (Naderializadeh et al., 2021) as our set encoder to leverage Sliced-Wasserstein geometric properties.

**Distributionally Robust Optimization (DRO).** DRO improves decision making by optimizing the worst-case expected objective over plausible distributions (Rahimian & Mehrotra, 2019; 2022; Bertsimas et al., 2019). A central theme of DRO is constructing ambiguity regions (Rahimian & Mehrotra, 2019; Wang et al., 2025; Nguyen et al., 2025; Ma et al., 2024; Yu et al., 2024; Inatsu et al., 2022). Classical formulations include moment-based and divergence-based approaches such as KL-divergence (Husain et al., 2023; Zhou et al., 2020; Namkoong & Duchi, 2016), while Staib & Jegelka (2019) utilizes kernel mean embeddings in Reproducing Kernel Hilbert Space and Sagawa et al. (2019) minimizes maximum expected loss over predefined groups. Ambiguity regions defined through Wasserstein distance have gained attention (Kuhn et al., 2019; Kwon et al., 2020). Wasserstein-DRO (WDRO) is appealing due to its geometric

interpretability, connections to regularization, and ability to capture distributional shifts (Gao et al., 2024; Micheli et al., 2025), with wide applications (Liu et al., 2025; Huang & Ding, 2025; Wu et al., 2025; Zhang et al., 2025). However, conventional WDRO requires modeling transport costs between individual elements, making direct SRL application computationally expensive. To address this concern, this work specifically proposes the SW-DRSO formulation that is tailored for modeling sets. By approximating Wasserstein distances through one-dimensional projections, SW-DRSO provides an efficient surrogate that preserves distributional robustness while reducing computational cost.

# 3. Preliminary and Motivation

## 3.1. Problem Description

**Set Representation Learning.** Let $\mathcal{X} \subseteq \mathbb{R}^d$ be the element space and $\mathcal{Y}$ the label space. Each input is an unordered set $S = \{x_i\}_{i=1}^n$ with variable cardinality $n$, where $x_i \in \mathcal{X}$. Set representation learning (SRL) learns a permutation-invariant and cardinality-agnostic encoder $f : \mathcal{S}(\mathcal{X}) \to \mathbb{R}^c$ that maps $S$ to a vector $v = f(S)$. The learned set representation $v$ is then used for downstream tasks, such as set classification (predicting label $y \in \mathcal{Y}$) or set matching via embedding-based ranking. We summarize all notations in Appendix A.

A common SRL training objective optimizes the *nominal* (source) risk via empirical risk minimization (ERM) (Zaheer et al., 2017; Lee et al., 2019). Let $\mathbb{P}_0$ be the nominal training distribution over clean sets. Conventional ERM solves:

$$\min_{f,g} \mathbb{E}_{(S,y)\sim\mathbb{P}_0}\big[\ell\big(g(f(S)), y\big)\big], \qquad (1)$$

where $g$ denotes the task-specific function (e.g., a classifier or a ranking function), operating on the set representation $v = f(S)$, and $\ell$ the corresponding loss.

**Set with Inference-Time Element Corruption.** We study a realistic scenario where training data is carefully curated, but sets observed at inference time suffer from *element-level corruption*. Given $S$ as the underlying clean set; at inference time, the model however may observe a corrupted version $S'$ in stead of $S$. Such corruptions are typically *sparse* and *heterogeneous*: only a subset of elements is affected, and corruption patterns vary across instances as well. Typical examples include missing elements and spurious outliers.

We assume the corruption is *label-preserving*: it only affects the partial set elements but not the ground-truth label of the holistic set semantic. Importantly, the learner does not observe which elements are corrupted at inference time; it only receives the full set $S'$.

## 3.2. Towards Learning Robust Set Representations

Minimizing Eq. (1) does not necessarily yield stable set representations under element-level corruption. In SRL, $f$ is typically implemented by aggregating element-wise features (e.g., pooling or attention) (Zaheer et al., 2017; Lee et al., 2019; Zhang et al., 2020), where a small fraction of corrupted elements may disproportionately affect the aggregated representation, even when the corruption is label-preserving. Consequently, the downstream predictor $g$, trained on clean embeddings, may fail when evaluated on corrupted embeddings at inference time.

**Robustness in SRL.** To fix this gap, a natural robustness principle is to optimize performance under the *worst* plausible corruptions. Let $\Gamma(S)$ denote the ambiguity region of plausible corruptions for each input $S$ (e.g., capturing sparse and heterogeneous element corruption at inference time). With $\mathbb{P}_0$ denotes the nominal data distribution over training inputs, one feasible solution is to leverage *Distributionally Robust Optimization (DRO)* (Scarf et al., 1957) as:

$$\min_{f,g} \mathbb{E}_{(S,y)\sim\mathbb{P}_0}\Big[\sup_{\mathbb{Q}\in\Gamma(S)} \mathbb{E}_{S'\sim\mathbb{Q}}\big[\ell\big(g(f(S')), y\big)\big]\Big]. \quad (2)$$

It can hedge against *unknown* inference-time corruption mechanisms by optimizing the worst-case risk (Rahimian & Mehrotra, 2019). However, for set-structured inputs, specifying $\Gamma(S)$ and solving the inner maximization can be prohibitive: for example, directly enumerating $\Gamma(S)$ over discrete set variants often leads to combinatorial explosion. These challenges motivate the development of a scalable DRO framework specifically for set representation learning.

# 4. Our Method

## 4.1. Overview

Our approach to learning robust set representations is twofold. ① We formulate sets as empirical measures and view corruptions as distributional perturbations. By characterizing the ambiguity region $\Gamma$ via the Sliced-Wasserstein (SW) distance, we derive the formulation of Sliced-Wasserstein Distributionally Robust Set Optimization (SW-DRSO). ② To mitigate the high computational cost, we further propose a synthesis-based approximation. This yields a *barycentric adversary* that is essentially a low-dimensional and differentiable parameterization to facilitate the efficient identification of target perturbations.

## 4.2. SW-DRSO Instantiation for Sets

### 4.2.1. MODELING SETS WITH SW METRIC

We model a set $S = \{x_i\}_{i=1}^n \subset \mathcal{X}$ as an empirical measure over the element space (Muandet et al., 2012; Zaheer et al.,

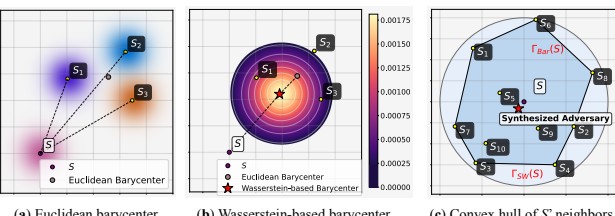

**(a)** Euclidean barycenter    **(b)** Wasserstein-based barycenter    **(c)** Convex hull of $S$' neighbors

*Figure 2.* Illustration of Wasserstein barycentric data synthesis.

2017; Edwards & Storkey, 2017):

$$\mu_S = \frac{1}{n}\sum_{i=1}^{n}\delta_{x_i} \in \mathcal{P}(\mathcal{X}), \qquad (3)$$

where $\delta_x$ is the Dirac measure at $x$ and $\mathcal{P}(\mathcal{X})$ denotes the set of probability measures on $\mathcal{X}$. This representation is permutation-invariant and explicitly captures the within-set distribution, so element corruption can be naturally viewed as a perturbation from $\mu_S$ to $\mu_{S'}$.

To quantify their discrepancy, we adopt the *2-Sliced-Wasserstein* distance (Kolouri et al., 2019), which is derived from the Optimal Transport (OT) theory (Kantorovich, 2006). For $\mu_S$ and $\mu_{S'}$ over $\mathbb{R}^d$, it is defined as:

$$SW(\mu_S, \mu_{S'}) = \left(\int_{\mathbb{S}^{d-1}} W(\mu_S^\omega, \mu_{S'}^\omega)^2 \, \mathrm{d}\omega\right)^{\frac{1}{2}}, \qquad (4)$$

where $\mathbb{S}^{d-1}$ is the unit sphere, and $\mu_S^\omega$, $\mu_{S'}^\omega$ denote the 1D projections under $\omega(x) = \omega^\top x$. $W$ denotes the standard 2-Wasserstein distance that can computed as:

$$W(\mu_S^\omega, \mu_{S'}^\omega) = \left(\inf_{\mathbf{p}\in \text{plans}(\mu_S^\omega, \mu_{S'}^\omega)} \int \|x - \mathbf{p}(x)\|^2 \mathrm{d}\mu_S^\omega(x)\right)^{\frac{1}{2}}. \qquad (5)$$

The infimum is taken over all transport plans between $\mu_S^\omega$ and $\mu_{S'}^\omega$. For 1D measures, a closed-form optimal solution $\mathbf{p}^+$ exists as the minimizer in Eq. (5).

### 4.2.2. **SW-DRSO INSTANTIATION**

Using SW to define feasible corruptions, we can try to instantiate $\Gamma(S)$ as an SW-ball around $\mu_S$:

$$\Gamma_{SW}(S) := \{S' : SW(\mu_S, \mu_{S'}) \leq \rho\}, \qquad (6)$$

where $\rho > 0$ controls the corruption radius. This yields SW-DRSO objective as follows:

$$\min_{\mathbf{f}, \mathbf{g}} \mathbb{E}_{(S,y)\sim \mathbb{P}_0}\left[\sup_{S'\in\Gamma_{SW}(S)} \ell(\mathbf{g}(\mathbf{f}(S')), y)\right]. \qquad (7)$$

A practical instantiation of Eq. (7) may expect a set encoder $\mathbf{f}(\cdot)$ whose representation respects the same SW geometry. Since standard pooling ignores this property (Skianis et al., 2020), we introduce a Wasserstein-aware encoder specifically aligned with SW-DRSO.

**Wasserstein-aware Set Encoder (Naderializadeh et al., 2021).** Following Naderializadeh et al. (2021), $\mathbf{f}(\cdot)$ can

be implemented with the following two stages:

- It first introduces a learnable reference set $O = \{o_h\}_{h=1}^H$ with empirical measure $\mu_O = \frac{1}{H}\sum_{h=1}^H \delta_{o_h}$. For each projection $\omega$, the optimal solution $\mathbf{p}^+$ is implemented as:

$$\mathbf{p}^+(t, \mu_S^\omega) = \left(\mathbf{F}_{\mu_S^\omega}\right)^{-1}\left(\mathbf{F}_{\mu_O^\omega}(t)\right), \quad t \in \mathbb{R}, \qquad (8)$$

where $\mathbf{F}$ and $\mathbf{F}^{-1}$ are the CDF and quantile function. Let $T_O^\omega = \texttt{sort}(\{\omega^\top o_h\}_{h=1}^H)$ and $T_S^\omega = \texttt{sort}(\{\omega^\top x_i\}_{i=1}^n)$. For empirical measures, $\mathbf{p}^+$ reduces to quantile matching between sorted scalars; Let $t_h^\omega$ be the $h$-th entry of $T_O^\omega$. When $H \neq n$, a simple index-based quantile approximation can be implemented:

$$\mathbf{p}^+(t_h^\omega, \mu_S^\omega) := T_S^\omega[q_h], \quad q_h := \left\lceil \frac{n}{H}h \right\rceil. \qquad (9)$$

- It then introduces $R$ Monte Carlo projections $\Omega = \{\omega_r\}_{r=1}^R$ to approximate the integral in Eq. (4). Therefore, we aggregate all OT coordinates by concatenation:

$$\mathbf{f}_{\texttt{SW}}(\mu_S) := \frac{1}{\sqrt{RH}}\texttt{Concat}_{r=1, h=1}^{R,H}\left(\mathbf{p}^+(t_h^{\omega_r}, \mu_S^{\omega_r})\right) \in \mathbb{R}^{RH}. \qquad (10)$$

### 4.3. Data Synthesis for SW-DRSO Approximation

Directly optimizing the inner supremum in Eq. (7) over the SW-ball is still intractable, as it requires searching over corrupted sets $S'$ that both satisfy the Sliced-Wasserstein constraint and maximize the downstream task loss after passing through $g$. Our goal is therefore to *approximately characterize* the most adversarial directions within an SW neighborhood while remaining consistent with ① the SW geometry that defines the corruption region and ② the task loss that determines worst-case behavior. To this end, as shown in Figure 2, we introduce a *barycentric synthesis* surrogate, which provides a *low-dimensional and differentiable* parameterization of SW-consistent perturbations, enabling efficient search for high-quality adversarial candidates. In practice, data synthesis has been used broadly to improve representation learning and model adaptation in many fields (He et al., 2026a;c; 2025).

### 4.3.1. **WASSERSTEIN BARYCENTER SYNTHESIS**

Let $v_S$ denote the set embedding from $\mathbf{f}_{\texttt{SW}}(\mu_S) \in \mathbb{R}^{RH}$. For each $S$ in a minibatch $\mathcal{B}$, we build a local neighbor pool $\mathcal{G}(S) = \{S_k\}_{k=1}^K \subseteq \mathcal{B}$ by $K$-NN under $\ell_2$ set embedding distance[2]. We denote the probability simplex by:

$$\Delta_K := \left\{\Lambda \in \mathbb{R}^K : \sum_{k=1}^K \lambda_k = 1 \text{ and } \lambda_k \geq 0\right\}. \qquad (11)$$

---

[2]In practice, we select either the $K$ nearest neighbors or all samples within Eculidean distance $\rho$, whichever is smaller. For notational simplicity, we denote the resulting set size as $K$.

Given $\Lambda = (\lambda_1, \dots, \lambda_K) \in \Delta_K$, we synthesize a perturbed embedding by convex mixing as follows:

$$\bar{v}_S(\Lambda) := \sum_{k=1}^{K} \lambda_k \, v_{S_k}. \tag{12}$$

Despite simplicity, Eq. (12) remains the set semantics within the SW-ball, as shown in Prop. 4.1 (all theoretical proofs are deferred to Appendix B).

**Proposition 4.1.** *For any $\Lambda \in \Delta_K$, let $S_\Lambda$ denote the implicit (virtual) set that corresponds to the synthesized embedding $\bar{v}_S(\Lambda)$. We have $S_\Lambda \in \Gamma(S)$.*

Moreover, this synthesis is *not* an arbitrary interpolation in representation space. Instead, it has a precise optimal-transport interpretation: under the 1D Wasserstein geometry, $\bar{v}_S(\Lambda)$ corresponds to the encoding of **Wasserstein barycenters**, i.e., the Fréchet mean that minimizes a weighted sum of squared $W_2$ distances on each projection.

**Proposition 4.2.** *Given reference measure $\mu_O$, for any batch pool $\mathcal{G}_S$ and weights $\Lambda \in \Delta_K$, the synthesized embedding $\bar{v}_S(\Lambda)$ corresponds to a (virtual) barycentric set, i.e., the SW Fréchet mean of all sets in $\mathcal{G}_S$:*

$$\bar{v}_S(\Lambda) = f_{\mathrm{SW}}(\mu_\Lambda), \mu_\Lambda \in \arg\min_{\mu \in \mathcal{P}(\mathcal{X})} \sum_{k=1}^{K} \lambda_k \, SW^2(\mu, \mu_{S_k}). \tag{13}$$

$\bar{v}_S(\Lambda)$ represents a most representative point within the local latent space, where the aggregate distance to all neighboring sets is minimized. This property enables us to restrict the adversary to barycentric perturbations without leaving the intended SW region, leading to a tractable surrogate of the original inner maximization as follows.

### 4.3.2. BARYCENTRIC ADVERSARY AS A SURROGATE

Based on the introduction of $\Delta_K$, we then reformulate the corruption region as the convex hull of embeddings:

$$\Gamma_{Bar}(S) := \big\{ \bar{v}_S(\Lambda) : \Lambda \in \Delta_K \big\}. \tag{14}$$

The resulting surrogate replaces the inner supremum with a low-dimensional maximization over $\Delta_K$:

$$\min_{f_{\mathrm{SW}}, g} \mathbb{E}_{(S,y)\sim\mathbb{P}_0} \Big[ \max_{\Lambda \in \Delta_K} \ell\big(g(\bar{v}_S(\Lambda)), y\big) \Big]. \tag{15}$$

With this construction, the barycentric inner maximization in Eq. (15) upper-bounds the discrete inner objective over individual sets. As a result, the adversary is strictly more expressive, yielding a more conservative surrogate that better captures worst-case behavior. This relationship is formalized in the following proposition:

**Proposition 4.3.** *Let $\mathcal{L}_{Disc}(S) := \max_{k\in[K]} \ell\big(g(v_{S_k}), y\big)$ and $\mathcal{L}_{Bar}(S) := \max_{\Lambda \in \Delta_K} \ell\big(g(\bar{v}_S(\Lambda)), y\big)$. For any predictor $g$ and loss $\ell$, we have:*

$$\mathcal{L}_{Disc}(S) \leq \mathcal{L}_{Bar}(S). \tag{16}$$

Intuitively, as shown in Figure 2(c), $\Gamma_{\mathrm{Bar}}(S)$ enlarges the

feasible adversarial set from a finite vertex set $\{v_{S_k}\}$ to its convex hull, hence the resulting inner maximization provides an upper bound of the discrete counterpart. We defer to Appendix B.4 a quantitative analysis, which characterizes the tightness of the barycentric surrogate by bounding $L_{\mathrm{Bar}}(S) - L_{\mathrm{Disc}}(S)$ under a local Lipschitz condition.

So far, we show that our barycentric synthesis ① *continuizes* the adversary beyond discrete samples while remaining major semantics within the local set neighborhood, and ② turns the intractable set-space search into an efficient low-dimensional maximization over $\Delta_K$.

### 4.4. Overall Objective and Optimization

In practice, we optimize a weighted combination of the nominal ERM loss and the barycentric robust term. Given $\alpha \geq 0$ as a trade-off hyperparameter, our overall training objective is formulated as:

$$\min_{f_{\mathrm{SW}}, g} \mathbb{E}_{(S,y)\sim\mathbb{P}_0} \Big[ \ell\big(g(v_S), y\big) + \alpha \cdot \max_{\Lambda \in \Delta_K} \ell\big(g(\bar{v}_S(\Lambda)), y\big) \Big], \tag{17}$$

For each training set $S$, we approximately solve the inner maximization in Eq. (17) using projected gradient ascent (Levitin & Polyak, 1966) on $\Delta_K$. We initialize the mixing weights with the uniform distribution $\Lambda^{(1)} = \big(\frac{1}{K}, \dots, \frac{1}{K}\big)$ and iteratively perform the following updates:

$$\tilde{\Lambda}^{(t+1)} = \Lambda^{(t)} + \eta \, \nabla_\Lambda \ell\big(g(\bar{v}_S(\Lambda^{(t)})), y\big), \tag{18}$$

$$\Lambda^{(t+1)} = \Pi_{\Delta_K}\left(\tilde{\Lambda}^{(t+1)}\right), \tag{19}$$

where $\eta > 0$ is the ascent step size and $\Pi_{\Delta_K}$ denotes the Euclidean projection onto $\Delta_K$ (enforcing $\lambda_k \geq 0$ and $\sum_{k=1}^{K} \lambda_k = 1$). After $T$ ascent steps, we obtain $\Lambda^\star = \Lambda^{(T)}$. We then fix $\Lambda^\star$ (by stopping gradients through $\Lambda^\star$) and update $f_{\mathrm{SW}}$ and $g$ by standard gradient descent on Eq. (17). We detail the pseudo-codes of our SW-DRSO in Appendix C and time complexity analysis in Appendix D.

## 5. Experiments

### 5.1. Experiment Setups

**Task Descriptions.** We validate the robustness of our method across four downstream tasks subject to varied inference-time corruptions. By encompassing domains ranging from 3D geometry to visual understanding, we expect a systematic evaluation under diverse data modalities.

- **Similar Set Ranking (Task I)**: At inference time, the *query set* is corrupted by noise (e.g., injected irrelevant elements and/or missing true elements). The goal is to retrieve and rank candidate sets that remain most similar to the underlying clean query, based on embedding distances. This task evaluates robustness of *set-to-set*

*similarity* modeling under noisy query composition.

- **Point Cloud Classification (Task II)**: The task classifies 3D objects represented as unordered point sets by aggregating local geometric information into a global representation. At inference time, the point cloud is corrupted by *inaccurate observations* (e.g., perturbed coordinates), while the input remains an unordered set of points. The model evaluates whether geometric aggregation is resilient to corrupted data.

- **Topic Set Expansion (Task III)**: The task expands a topic by selecting relevant keywords from a vocabulary given a seed keyword set, assessing semantic coherence modeling at the set level. At inference time, the *seed keyword set* is imperfect, containing missing topic-defining words and/or noisy, off-topic words. We assess robustness of semantic set modeling under such conditions.

- **Patch-set Visual Recognition (Task IV)**: The task aims to perform visual classification by encoding sets of local patch elements. At inference time, element corruption is introduced at the *patch level*, where a subset of patches is masked out or perturbed by additive noise. The model is therefore required to perform robust visual recognition against localized corruptions.

**Data Descriptions.** We adopted a standardized data partitioning protocol across all tasks: datasets were split into training and testing subsets with an 80:20 ratio. The training portion was subdivided (80:20) to create a validation set for hyperparameter tuning. To evaluate robustness under inference-time corruption, datasets are grouped with different corruption levels. Specifically, only the validation and test sets are partitioned into *clean/mild/severe* subsets. Mild and severe parts introduce corruption to approximately 10% and 40% of the input elements or regions. This protocol trains models on clean data but evaluates them under varying degrees of corrupted inputs, reflecting real deployment conditions. Data statistics are reported in Appendix E.1.

**Baselines.** We compare against a diverse suite of set representation baselines, including classic pooling (**MeanP/MaxP**) (Lin et al., 2013), universal set-function modeling (**DeepSet**) (Zaheer et al., 2017), prototype matching via OT theory (**RepSet**) (Skianis et al., 2020), attention-based set Transformers (**SetTRSM**) (Lee et al., 2019), information-theoretic learning (**DIEM**) (Kim, 2022), learnable sorting-based pooling (**FSPool**) (Zhang et al., 2020), and recent strong OT-based methods such as **PSWE** (Naderializadeh et al., 2021) and frequency-domain multiset modeling (**FSW**) (Amir & Dym, 2025). Detailed descriptions of these baselines are provided in the Appendix E.2.

**Configurations.** Optimization for all other models is performed using the default Adam optimizer (Kingma & Ba,

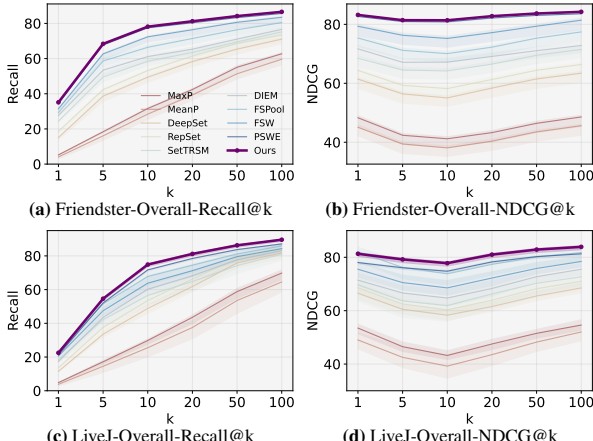

*Figure 3.* Task I Performance comparison in terms of Recall@k and NDCG@k. Solid curves correspond to the overall performance and the shaded region indicates the range bounded by the upper clean split and the lower severe split.

2015), with the learning rate tuned via grid search within the set $\{10^{-4}, 10^{-3}, 10^{-2}\}$. Regarding baseline configurations, we strictly followed the hyperparameter settings reported in their original literature; for models without specified configurations, optimal settings were determined via grid search. The implementation was carried out in PyTorch 1.13.0 (Python 3.9) on a Linux server equipped with eight NVIDIA L40S GPUs. To account for statistical variance, all reported results represent the mean performance and standard deviations over five independent runs. Comprehensive details for reproducibility are provided in Appendix F.

### 5.2. Experimental Results

#### 5.2.1. TASK I EVALUATION

For the Similar Set Ranking task, we evaluate on two large-scale real-world social network datasets, *Friendster* (Yang & Leskovec, 2015) and *LIVEJ* (Mislove et al.). Figure 3 summarizes the results under different corruption conditions. Sets are ranked by measuring the Euclidean distance in the learned embedding space, with Recall@k and NDCG adopted as the evaluation metrics, which are widely used in retrieval and recommendation settings (He & Ma, 2024; Chen et al., 2023a; Luo et al., 2025b; 2026d; Chen et al., 2022a;b; Cui et al., 2024; 2026; Zhang et al., 2026b).

We have two major observations: ① First, our method in purple consistently achieves strong performance across all datasets and metrics, demonstrating clear overall effectiveness. ② Second, our method exhibits the narrowest shaded band among all compared methods, meaning the performance gap between the severe and clean splits is the smallest. This indicates that our approach is more robust to the inference-time corruption, maintaining stable ranking qual-

*Table 1.* Task II performance comparison (%). Best and second-best results are shown in **bold** and underlined.

| Method | Overall | Clean | Mild | Severe |
|--------|---------|-------|------|--------|
| MeanP | 80.41±0.79 | 84.85±0.73 | 83.96±1.18 | 63.97±3.03 |
| MaxP | 77.29±0.73 | 85.49±0.70 | 79.10±0.98 | 54.08±2.82 |
| DeePSet | 78.67±0.74 | 85.80±0.67 | 80.83±0.99 | 57.60±2.92 |
| RepSet | 79.55±0.76 | 85.09±0.69 | 82.93±1.06 | 60.63±2.97 |
| SetTRSM | 75.28±0.65 | 86.17±0.48 | 75.43±0.85 | 47.81±2.76 |
| DIEM | 80.59±0.68 | 85.10±0.90 | 83.80±0.95 | 64.50±2.10 |
| FSPool | 80.16±0.83 | 85.24±1.41 | 83.51±0.78 | 62.43±1.83 |
| FSW | 81.14±0.74 | 83.84±0.59 | 84.62±1.12 | 69.19±2.93 |
| PSWE | 81.50±0.70 | 84.51±0.49 | 84.38±1.08 | 69.65±2.83 |
| **Ours** | **82.86*** **±0.71** | **86.32±0.78** | **85.68*** **±0.73** | **69.99*** **±2.78** |

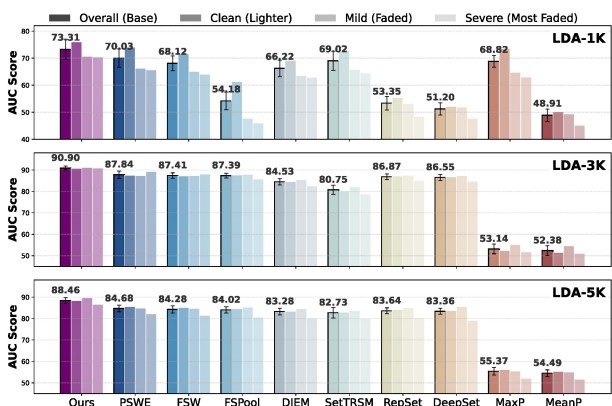

*Figure 4.* Task III performance comparison. For each method, the leftmost black-bordered bar represents the overall score, with exact values annotated on top. The symbol "I" measures the difference between the clean and severe performances.

*Table 2.* Task IV performance comparison (%).

| Method | Overall | Clean | Mild | Severe |
|--------|---------|-------|------|--------|
| MaxP | 48.27±0.55 | 55.43±0.67 | 50.18±0.84 | 27.53±1.08 |
| MeanP | 49.99±0.58 | 56.83±0.42 | 52.28±0.64 | 29.46±0.83 |
| DeepSet | 48.49±0.85 | 56.12±0.65 | 51.29±0.91 | 25.21±1.18 |
| RepSet | 49.52±0.59 | 57.19±0.61 | 51.82±0.78 | 26.88±0.96 |
| SetTRSM | 50.26±0.71 | 59.48±0.54 | 53.04±0.73 | 23.06±1.24 |
| DIEM | 51.91±0.48 | 59.42±0.71 | 53.67±0.85 | 30.51±1.09 |
| FSPool | **53.13±0.49** | **61.88±0.69** | 54.15±0.92 | 29.74±1.31 |
| FSW | 52.15±0.65 | 59.85±0.55 | 53.45±0.78 | 30.95±0.95 |
| PSWE | 51.21±0.42 | 57.76±0.27 | 53.49±0.53 | 31.43±0.63 |
| **Ours** | 53.03±0.89 | 60.33±1.05 | **54.95*** **±1.19** | **31.89*** **±0.79** |

ity even under challenging conditions. Complete numerical results are provided in Appendix G.

### 5.2.2. TASK II EVALUATION

We evaluate point cloud classification on the standard ModelNet benchmark (Wu et al., 2015) with ISAB as the feature backbone (Lee et al., 2019), with accuracy results summarized in Table 1. ① Our method achieves the best performance across all evaluation settings over the strongest baseline in terms of overall accuracy. This demonstrates the effectiveness of our approach for learning robust set representations in point cloud data classification. ② When examining different corruption levels, we observe that existing methods exhibit varying robustness performance. In particular, PSWE shows relatively strong resistance under severe corruption, achieving the best baseline performance in this regime, while FSW performs competitively under mild corruption, ranking as the second-best baseline. These results suggest that different aggregation mechanisms emphasize different aspects of robustness. Nevertheless, our method consistently outperforms all baselines under clean, mild, and severe conditions, indicating improved stability across a wide range of inference-time corruptions. ③ To assess the statistical reliability of these improvements, we conduct the Wilcoxon signed-rank significance test across all experimental settings. The results show that the performance gains of our method are statistically significant at the 95% confidence level for the vast majority of comparisons.

### 5.2.3. TASK III EVALUATION

We conduct evaluations on three benchmark datasets, *LDA-1k*, *LDA-3k*, and *LDA-5k* (Zaheer et al., 2017), which differ in both scale and topical coverage. Tokens are embedded via word2vec (Mikolov, 2013) and scored by semantic similarity to the query, with ranking quality measured by AUC. As shown in Figure 4, ① our method consistently achieves the best overall AUC performance across all three datasets. Notably, this advantage is driven by strong performance under the clean data partitions, indicating that our approach effectively captures the core semantic structure required for

topic expansion. ② Beyond overall accuracy, our method also demonstrates superior robustness to corruption. Across all datasets, the performance gap between the clean and severe conditions is relatively small compared to competing methods, as indicated by the I-bar markers. This effect is particularly pronounced on *LDA-3k*, where our method exhibits the smallest clean–severe discrepancy, suggesting strong resilience to inference-time corruption. On *LDA-1k* and *LDA-5k*, although the gap slightly increases, our approach remains highly competitive and consistent performance. Detailed results are reported in Appendix H.

### 5.2.4. TASK IV EVALUATION

For Task IV, we evaluate on the large-scale NWPU-RESISC45 dataset (Cheng et al., 2017). We use the same MLP-based patch encoder in (Naderializadeh et al., 2021) and report classification accuracy as the evaluation metric. ① In general, our model attains highly competitive overall accuracy and is effectively on par with the best-performing baseline. ② Although our method ranks second on the clean subset, it outperforms most competitors on the mild and severe subsets. This indicates improved robustness: the model maintains higher accuracy under more challenging conditions, showing that the proposed method achieves

*Table 3.* Ablation study on LiveJ dataset.

| W/o | Recall@10 | | | | NDCG@10 | | | |
|---|---|---|---|---|---|---|---|---|
| | Overall | Clean | Mild | Severe | Overall | Clean | Mild | Severe |
| SW-DRSO | 72.58 | 73.45 | 72.22 | 70.96 | 74.40 | 75.23 | 74.14 | 72.69 |
| $f_{sw}$ | 69.55 | 70.23 | 69.17 | 68.43 | 70.77 | 71.38 | 70.77 | 69.23 |
| WBS | 72.92 | 73.81 | 72.63 | 71.13 | 75.68 | 76.24 | 75.40 | 74.72 |
| **Ours** | **74.86** | **75.64** | **74.63** | **73.25** | **77.80** | **78.49** | **77.54** | **76.44** |

*Table 4.* Empirical study of our approximation strategies. Time represents the training time per epoch.

| Method | Overall | Clean | Mild | Severe | Time (min) |
|---|---|---|---|---|---|
| Sinkhorn | 74.81 | 75.52 | **74.71** | 73.20 | 21.5 |
| Disc-Adversary | 74.45 | 75.18 | 74.01 | 73.27 | 18.4 |
| **Ours** | **74.86** | **75.64** | 74.63 | **73.25** | **15.6** |

state-of-the-art accuracy under varying test conditions.

### 5.3. Empirical Analysis

We evaluate our framework's design choices using the LiveJ dataset (introduced in Task 1).

#### 5.3.1. MODULE-WISE ABLATION STUDY

We construct three ablation variants against the full model. The comparison results are presented in Table 3.

- `w/o SW-DRSO`: We eliminate our proposed SW-DRSO framework entirely. Instead, the model is directly optimized using standard Empirical Risk Minimization in Eq. (17). We notice that, the performance drop, particularly in the *severe* corruption setting, indicates the critical role of our SW-DRSO frame in enhancing robustness.

- `w/o` $f_{sw}$: We replace the proposed Sliced-Wasserstein-based set encoding mechanism with a rudimentary mean pooling strategy to assess the efficacy of the encoding scheme. This variant exhibits the most substantial degradation across all metrics. This result empirically verifies that the Wasserstein-based geometric awareness can capture complex set structures, which is more effective than simple aggregation strategy.

- `w/o WBS`: We disable the Wasserstein Barycenter Synthesis module. Instead of synthesizing barycenters, we utilize set embeddings randomly sampled from the current batch as the optimization targets. The decline in performance suggests that our synthesized barycenters can effectively provide supervision signals compared to random in-batch samples.

#### 5.3.2. STUDY OF OUR APPROXIMATION STRATEGIES

Our framework employs a twofold approximation strategies: We first adopt Sliced-Wasserstein metric for tractable

Wasserstein geometry, and then we propose *barycentric adversary* as a differentiable surrogate for the inner maximization in DRO. We thus conduct empirical studies to validate the effectiveness of these two designs.

**Wasserstein Geometry Approximation.** To evaluate the efficacy of our approximation, we substitute the SW metric with the standard Wasserstein metric. Given that the computational complexity of the original Wasserstein distance scales at $O(n^3 \log n)$ or higher (Pele & Werman, 2009), we adopt the Sinkhorn computation algorithm as a common practice (Cuturi, 2013). Consequently, we implement a Sinkhorn-based variant for both set encoding and the SW-DRSO formulation. As shown in Table 4, while the `Sinkhorn` baseline achieves competitive overall performance, our SW-based method still outperforms it with significantly improved computational efficiency, i.e., 21.5 min vs. 15.6 min per epoch (27% faster). This demonstrates that SW approximation provides a better trade-off between accuracy and efficiency for large-scale settings.

**Barycentric Adversary for Optimization.** The second approximation lies within the optimization process, where we employ a barycentric adversary as a surrogate. We compare our approach against the *Discrete Adversary* strategy (corresponding to $\mathcal{L}_{Disc}$ in Proposition 4.3), which searches over all discrete sets in $\mathcal{G}$ to identify the worst-case instance for gradient estimation. As shown in Table 4, our method outperforms `Disc-Adversary`, empirically validating Proposition 4.3, while achieving better training efficiency.

**Comparison with Random Combinatorial Search.** To further examine whether an explicit search over corrupted set candidates can replace the barycentric adversary, we evaluate random combinatorial sampling (RCS) on the LiveJ dataset. RCS samples candidate corrupted sets in the discrete set space for $m$ rounds and uses high-loss candidates for robust training. We also evaluate an SW-constrained variant that filters sampled candidates by the local Sliced-Wasserstein distance. As shown in Table 5, increasing the number of sampled candidates substantially increases training time, while the performance gain remains limited. In contrast, our barycentric adversary does not require a sampling-round parameter and achieves 74.86 Recall@10 with 15.6 min per epoch, yielding a better accuracy-efficiency trade-off by optimizing continuously over simplex weights rather than sampling discrete corrupted sets.

**Cardinality-wise Scalability.** We further evaluate scalability by grouping LiveJ data instances according to the input set cardinality, i.e., the number of elements contained in each input set. Each cardinality group corresponds to a range of set sizes, such as sets with 1–10 elements or more than 30 elements. As shown in Table 7, SW-DRSO

*Table 5.* Random combinatorial search analysis on LiveJ. R@10 denotes Recall@10, $m$ denotes the number of sampling rounds, and time is measured per epoch.

| Method | $m = 2$ | | $m = 5$ | | $m = 10$ | |
|---|---|---|---|---|---|---|
| | R@10 | Time | R@10 | Time | R@10 | Time |
| RCS | 69.45 | 21.7 | 70.56 | 23.5 | 71.13 | 27.4 |
| RCS w/ SW | 69.52 | 22.6 | 71.81 | 24.8 | 71.74 | 28.1 |

*Table 6.* Comparison with other DRO variants.

| Method | Overall | Clean | Mild | Severe | Time (min) |
|---|---|---|---|---|---|
| WDRO | 74.72 | 75.32 | 74.21 | 73.10 | 17.5 |
| KL-DRO | 74.45 | 74.92 | 74.14 | 73.87 | **14.2** |
| MMD-DRO | 74.09 | 74.86 | 73.60 | 73.07 | 18.7 |
| **Ours** | **74.86** | **75.64** | **74.63** | **73.25** | 15.6 |

maintains consistent Recall@10 gains across all cardinality groups, while its training time grows moderately with set size. This suggests that the barycentric inner optimization introduces manageable overhead beyond the base SW encoder.

### 5.3.3. COMPARISON WITH OTHER DRO VARIANTS

We benchmark SW-DRSO against classical DRO variants, including WDRO (Gao et al., 2024), KL-DRO (Namkoong & Duchi, 2016), and MMD-DRO (Staib & Jegelka, 2019). Detailed descriptions are provided in Appendix I.1. We implement them on our encoder architecture by replacing the distance metric used in the local neighbor pool construction (Section 4.3.1) with their respective divergence measures. As shown in Table 6, our SW-DRSO performs the best while maintaining competitive training efficiency. WDRO and MMD-DRO suffer from significant computational overhead while delivering inferior accuracy. We attribute this to the geometric alignment between our Wasserstein-based DRO formulation and Sliced-Wasserstein encoder, which enables effective robust optimization, unlike the geometric mismatches in other variants.

## 6. Conclusion

In this work, we addressed the challenge of Inference-time Element Corruption in Set Representation Learning, an issue where deployed models face sparse degradations typically absent from training data. We proposed SW-DRSO, a scalable distributionally robust optimization framework that leverages Sliced-Wasserstein geometry to define a tractable ambiguity region. By introducing a barycentric adversary, we transformed the computationally prohibitive search for worst-case perturbations into an efficient optimization process. Extensive experiments across diverse tasks and data modalities demonstrate that SW-DRSO effectively enhances robustness against severe corruption while maintaining competitive accuracy on clean inputs. A promising future direction is to extend such robust set optimization to LLM- and agent-centric settings where robustness is increasingly re-

*Table 7.* Cardinality-wise scalability on LiveJ. Cardinality groups are defined by the number of elements in each input set. R@10 denotes Recall@10, and time is measured per epoch.

| Method | $[1, 10]$ | | $[11, 20]$ | | $[20, 30]$ | | $> 30$ | |
|---|---|---|---|---|---|---|---|---|
| | R@10 | Time | R@10 | Time | R@10 | Time | R@10 | Time |
| FSW | 64.24 | 11.7 | 64.18 | 11.9 | 64.32 | 12.2 | 64.30 | 12.7 |
| PSWE | 71.44 | 12.1 | 71.81 | 12.6 | 71.76 | 13.2 | 71.92 | 13.9 |
| **Ours** | **74.65** | 14.6 | **74.91** | 15.9 | **74.77** | 16.7 | **74.81** | 17.1 |

quired under distributional changes (He et al., 2026b; Zhang et al., 2026a; Wu et al., 2026; Huang et al., 2025; Chen et al., 2026a; Luo et al., 2026b;c).

## Acknowledgements

This work is supported in part by NSF under grants III-2106758, and POSE-2346158.

## Impact Statement

This work addresses a practical reliability gap in Set Representation Learning: element corruptions at inference time may disproportionately distort pooled set embeddings and cause downstream failures. To mitigate this, we propose a scalable distributionally robust optimization framework that represents sets as empirical measures and defines tractable corruption regions, enabling robustness without discrete enumeration of corrupted set variants. We further introduce a barycentric data synthesis strategy that turns the otherwise difficult inner worst-case search into efficient parameterization optimization. Based on our proposed method, we aim to improve the operational robustness of set-based models, reducing failure rates when inputs are incomplete, noisy, or adversarially perturbed.

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

# A. Notation Explanations

All notations are summarized and explained in Table 8.

*Table 8.* Summary of notation used throughout the paper.

| Symbol | Description |
|---|---|
| $\mathcal{X} \subseteq \mathbb{R}^d, \mathcal{Y}$ | Element space of individual set elements and label space |
| $x_i \in \mathcal{X}$ | The $i$-th element in a set |
| $S = \{x_i\}_{i=1}^n, S'$ | Unordered input set with cardinality $n$ and corrupted version of set $S$ at inference time |
| $\mathcal{S}(\mathcal{X})$ | Space of all finite subsets of $\mathcal{X}$ |
| $f : \mathcal{S}(\mathcal{X}) \to \mathbb{R}^c$ | Permutation-invariant set encoder |
| $v_S = f(S)$ | Learned representation (embedding) of set $S$ |
| $g(\cdot)$ | Task-specific predictor (e.g., classifier or ranking function) |
| $\ell(\cdot, \cdot)$ | Task loss function |
| $P_0$ | Nominal training distribution over clean sets |
| $\Gamma(S)$ | Ambiguity (corruption) region associated with set $S$ |
| $\Gamma_{\mathrm{SW}}(S)$ | Sliced-Wasserstein ambiguity region centered at $S$ |
| $\Gamma_{\mathrm{Bar}}(S)$ | Barycentric surrogate ambiguity region |
| $\rho$ | Radius of the ambiguity region |
| $\mu_S$ | Empirical measure associated with set $S$ |
| $\delta_x$ | Dirac measure at location $x$ |
| $\mathcal{P}(\mathcal{X})$ | Space of probability measures over $\mathcal{X}$ |
| $\mu_{S'}$ | Empirical measure of corrupted set $S'$ |
| $\mathbb{S}^{d-1}$ | Unit sphere in $\mathbb{R}^d$ |
| $\omega \in \mathbb{S}^{d-1}$ | Projection direction |
| $\omega(x) = \omega^\top x$ | One-dimensional projection of element $x$ |
| $\mu_S^\omega$ | Projected 1D measure of $\mu_S$ along $\omega$ |
| $W(\cdot, \cdot)$ | 2-Wasserstein distance |
| $\mathrm{SW}(\mu_S, \mu_{S'})$ | Sliced-Wasserstein distance between measures |
| $O = \{o_h\}_{h=1}^H$ | Learnable reference set |
| $H, R$ | Cardinality of the reference set and number of Monte Carlo projection directions |
| $\mu_O$ | Empirical measure of the reference set |
| $F_\mu, F_\mu^{-1}$ | Cumulative distribution function (CDF) of measure $\mu$ and quantile function of measure $\mu$ |
| $p^+(\cdot)$ | Optimal 1D Wasserstein transport map |
| $\Omega = \{\omega_r\}_{r=1}^R$ | Set of projection directions |
| $f_{\mathrm{SW}}(S)$ | Sliced-Wasserstein-based set embedding |
| $\mathcal{B}$ | Mini-batch of training sets |
| $\mathcal{G}(S)$ | Local neighbor pool of set $S$ |
| $K$ | Number of neighbors in $\mathcal{G}(S)$ |
| $S_k$ | The $k$-th neighboring set of $S$ |
| $v_{S_k}$ | Embedding of neighbor set $S_k$ |
| $\Delta_K$ | Probability simplex in $\mathbb{R}^K$ |
| $\Lambda = (\lambda_1, \ldots, \lambda_K)$ | Barycentric mixing weights |
| $\bar{v}_S(\Lambda)$ | Synthesized barycentric embedding of set $S$ |
| $S_\Lambda$ | Implicit (virtual) barycentric set induced by $\Lambda$ |
| $\eta, \alpha$ | Step size for projected gradient ascent, and trade-off coefficient between ERM and robust loss |
| $T$ | Number of inner maximization steps |
| $\Pi_{\Delta_K}(\cdot)$ | Euclidean projection onto simplex $\Delta_K$ |
| $\Lambda^{(t)}, \Lambda^*$ | Mixing weights at iteration $t$ and final adversarial barycentric weights |

## B. Theoretical Analysis

### B.1. Proof of Proposition 4.1

*Proof.* Let $\delta_k := v_{S_k} - v_S$. Since $\bar{v}_S(\Lambda) - v_S = \sum_{k=1}^{K} \lambda_k \delta_k$ and the function $\phi(x) = \|x\|_2^2$ is convex, Jensen's inequality directly gives:

$$\left\| \bar{v}_S(\Lambda) - v_S \right\|_2^2 = \left\| \sum_{k=1}^{K} \lambda_k \delta_k \right\|_2^2 \leq \sum_{k=1}^{K} \lambda_k \|\delta_k\|_2^2. \tag{1}$$

Since $\|\delta_k\|_2^2 = \|v_{S_k} - v_S\|_2^2 \leq \rho^2$ for all $k$ (the selection principle of $\mathcal{G}$) and $\sum_k \lambda_k = 1$, we obtain

$$\sum_{k=1}^{K} \lambda_k \|\delta_k\|_2^2 \leq \sum_{k=1}^{K} \lambda_k \rho^2 = \rho^2. \tag{2}$$

Furthermore, according to (Naderializadeh et al., 2021), we have:

$$\|\bar{v}_S(\Lambda) - v_S\|_2 = SW(\mu_{S(\Lambda)}, \mu_S). \tag{3}$$

We provide the derivation of Eq. (3) for completeness in Appendix B.5. This implies that:

$$SW(\mu_{S(\Lambda)}, \mu_S) \leq \rho, \tag{4}$$

which completes the proof. □

### B.2. Proof of Proposition 4.2

*Proof.* For each direction $\omega$, the 1D Wasserstein barycenter is formulated as:

$$\mu_\Lambda^\omega \in \arg \min_{\mu \in \mathcal{P}(\mathbb{R})} \sum_{k=1}^{K} \lambda_k W_2^2(\mu, \mu_{S_k}^\omega). \tag{5}$$

Then by optimality of $\mu_\Lambda^\omega$, for any candidate $\mu$ and every $\omega$, we have:

$$\sum_{k=1}^{K} \lambda_k W_2^2(\mu^\omega, \mu_{S_k}^\omega) \geq \sum_{k=1}^{K} \lambda_k W_2^2(\mu_\Lambda^\omega, \mu_{S_k}^\omega). \tag{6}$$

Considering that, (as shown in Eq. (4)),

$$SW^2(\mu, \nu) = \int_{\mathbb{S}^{d-1}} W_2^2(\mu^\omega, \nu^\omega) \, d\omega = \mathbb{E}[W_2^2(\mu^\omega, \nu^\omega)], \tag{7}$$

by integrating over $\omega$, we then have:

$$\sum_{k=1}^{K} \lambda_k SW^2(\mu, \mu_{S_k}) \geq \sum_{k=1}^{K} \lambda_k SW^2(\mu_\Lambda, \mu_{S_k}). \tag{8}$$

This implies that $\mu_\Lambda$ is an SW Fréchet mean:

$$\mu_\Lambda \in \arg \min_{\mu \in \mathcal{P}(\mathcal{X})} \sum_{k=1}^{K} \lambda_k SW^2(\mu, \mu_{S_k}). \tag{9}$$

Finally, we relate $\mu_\Lambda$ to the synthesized embedding. We slightly abuse notation by letting $\mathtt{f}_{SW}$ act on a measure through its slice-wise reference-to-input OT coordinates. Then given any slice $\omega$ and reference coordinate $t$, let $u := F_{\mu_O^\omega}(t) \in (0, 1)$. In 1D, the $W_2$ barycenter has the quantile characterization (Agueh & Carlier, 2011; Peyré & Cuturi, 2019):

$$F_{\mu_\Lambda^\omega}^{-1}(u) = \sum_{k=1}^{K} \lambda_k F_{\mu_{S_k}^\omega}^{-1}(u). \tag{10}$$

Hence, the reference-to-input OT coordinate $\mathrm{p}^+(t, \nu) = F_\nu^{-1}(F_{\mu_O^\omega}(t)) = F_\nu^{-1}(u)$ is linear in $\Lambda$:

$$\mathrm{p}^+(t, \mu_\Lambda^\omega) = \sum_{k=1}^{K} \lambda_k \, \mathrm{p}^+(t, \mu_{S_k}^\omega). \tag{11}$$

Concatenating Eq. (11) over the sampled slices as in $\mathrm{f}_{\mathrm{SW}}$ finally yields:

$$\mathrm{f}_{\mathrm{SW}}(\mu_\Lambda) = \sum_{k=1}^{K} \lambda_k \, \mathrm{f}_{\mathrm{SW}}(\mu_{S_k}) = \sum_{k=1}^{K} \lambda_k \, v_{S_k} = \bar{v}_S(\Lambda), \tag{12}$$

which completes the proof. $\qquad\square$

### B.3. Proof of Proposition 4.3

*Proof.* Recall that the barycentric region is defined as the convex hull of the neighbor embeddings:

$$\Gamma_{\mathrm{Bar}}(S) := \left\{ \bar{v}_S(\Lambda) = \sum_{k=1}^{K} \lambda_k v_{S_k} : \Lambda \in \Delta_K \right\}, \tag{13}$$

where $\Delta_K = \{\lambda \in \mathbb{R}^K : \sum_{k=1}^{K} \lambda_k = 1, \lambda_k \geq 0\}$ is the probability simplex. The discrete and barycentric inner objectives are defined as $L_{\mathrm{Disc}}(S) := \max_{k \in [K]} \ell\big(g(v_{S_k}), y\big)$, and $L_{\mathrm{Bar}}(S) := \max_{v \in \Gamma_{\mathrm{Bar}}(S)} \ell\big(g(v), y\big)$. Since $\Gamma_{\mathrm{Bar}}(S)$ is the convex hull of $\{v_{S_k}\}_{k=1}^{K}$, each vertex $v_{S_k}$ itself is contained in $\Gamma_{\mathrm{Bar}}(S)$ by choosing $\Lambda$ such that $\lambda_k = 1$ and $\lambda_j = 0$ for all $j \neq k$, we recover $\bar{v}_S(\Lambda) = v_{S_k}$. Therefore, we have: $\{v_{S_k}\}_{k=1}^{K} \subseteq \Gamma_{\mathrm{Bar}}(S)$.

Then maximizing the same function $\phi(v) := \ell\big(g(v), y\big)$ over a superset cannot decrease the optimum, hence we have:

$$\max_{k \in [K]} \phi(v_{S_k}) \leq \max_{v \in \Gamma_{\mathrm{Bar}}(S)} \phi(v). \tag{14}$$

That is, $L_{\mathrm{Disc}}(S) \leq L_{\mathrm{Bar}}(S)$, which proves the claim. $\qquad\square$

### B.4. Quantitative Gap Between $L_{\mathrm{Bar}}(S)$ and $L_{\mathrm{Disc}}(S)$

**Proposition B.1.** *Let $\phi(z) = \ell(g(z), y)$. We define the discrete and barycentric inner objectives as:*

$$L_{\mathrm{Disc}}(S) := \max_{k \in [K]} \phi(v_{S_k}), \quad L_{\mathrm{Bar}}(S) := \max_{z \in \Gamma_{\mathrm{Bar}}(S)} \phi(z). \tag{15}$$

*Assume $\phi$ is $L$-Lipschitz on $\Gamma_{\mathrm{Bar}}(S)$. Then the following quantitative bound holds:*

$$0 \leq L_{\mathrm{Bar}}(S) - L_{\mathrm{Disc}}(S) \leq 2L \cdot \rho. \tag{16}$$

*Proof.* We have prove the first inequality in Appendix B.3. For many standard loss functions and predictors, $\phi$ is locally Lipschitz on bounded (hence compact) domains (Shalev-Shwartz & Ben-David, 2014; Bartlett et al., 2017). If $\phi$ is $L$-Lipschitz on $\Gamma_{\mathrm{Bar}}(S)$ ($L$ is a constant), then for all $z, z' \in \Gamma_{\mathrm{Bar}}(S)$, we have:

$$|\phi(z) - \phi(z')| \leq L \cdot \|z - z'\|_2. \tag{17}$$

Let $z^\star \in \arg\max_{z \in \Gamma_{\mathrm{Bar}}(S)} \phi(z)$ so that $L_{\mathrm{Bar}}(S) = \phi(z^\star)$. Choose $k^\star \in \arg\min_{k \in [K]} \|z^\star - v_{S_k}\|_2$. Then we have

$$|\phi(z^\star) - \phi(v_{S_{k^\star}})| \leq L \cdot \|z^\star - v_{S_{k^\star}}\|_2, \tag{18}$$

which implies that

$$\phi(z^\star) - \phi(v_{S_{k^\star}}) \leq |\phi(z^\star) - \phi(v_{S_{k^\star}})| \leq L \cdot \|z^\star - v_{S_{k^\star}}\|_2. \tag{19}$$

Moreover, $\phi(v_{S_{k^\star}}) \leq \max_{k \in [K]} \phi(v_{S_k}) = L_{\mathrm{Disc}}(S)$. Therefore, we have:

$$\begin{aligned}
L_{\mathrm{Bar}}(S) - L_{\mathrm{Disc}}(S) &= \phi(z^\star) - L_{\mathrm{Disc}}(S) \\
&\leq \phi(z^\star) - \phi(v_{S_{k^\star}}) \\
&\leq L \min_{k \in [K]} \|z^\star - v_{S_k}\|_2 \\
&\leq L \sup_{z \in \Gamma_{\mathrm{Bar}}(S)} \min_{k \in [K]} \|z - v_{S_k}\|_2.
\end{aligned} \tag{20}$$

Then based on the definition of $G(S)$, we have

$$\|v_{S_k} - v_S\|_2 \leq \rho, \qquad \forall S_k \in G(S). \tag{21}$$

In addition, for any $z \in \Gamma_{\mathrm{Bar}}(S)$, there exists $\Lambda \in \Delta_K$ such that $z = \sum_{k=1}^{K} \lambda_k v_{S_k}$. Using convexity of $\|\cdot\|_2^2$ (or Jensen's inequality) together with (21), we obtain

$$
\begin{aligned}
\|z - v_S\|_2^2 = \left\| \sum_{k=1}^{K} \lambda_k (v_{S_k} - v_S) \right\|_2^2 &\leq \sum_{k=1}^{K} \lambda_k \|v_{S_k} - v_S\|_2^2 \\
&\leq \sum_{k=1}^{K} \lambda_k \rho^2 = \rho^2,
\end{aligned}
\tag{22}
$$

hence $\|z - v_S\|_2 \leq \rho$ for all $z \in \Gamma_{\mathrm{Bar}}(S)$. Finally, by the triangle inequality, for any $z \in \Gamma_{\mathrm{Bar}}(S)$ and any $k \in [K]$,

$$\|z - v_{S_k}\|_2 \leq \|z - v_S\|_2 + \|v_S - v_{S_k}\|_2 \leq \rho + \rho = 2\rho. \tag{23}$$

Therefore, $\min_{k \in [K]} \|z - v_{S_k}\|_2 \leq 2\rho$ that leads to:

$$\sup_{z \in \Gamma_{\mathrm{Bar}}(S)} \min_{k \in [K]} \|z - v_{S_k}\|_2 \leq 2\rho. \tag{24}$$

Plugging this into Eq. (20) finally completes the proof. $\qquad\square$

**Proposition B.2.** *Let $V_S := \{v_{S_k}\}_{k=1}^{K}$ denote the embeddings of $G(S)$. Let $\phi(z) = \ell(g(z), y)$. We consider the discrete and barycentric inner objectives:*

$$L_{\mathrm{Disc}}(S) := \max_{k \in [K]} \phi(v_{S_k}), \quad L_{\mathrm{Bar}}(S) := \max_{z \in \Gamma_{\mathrm{Bar}}(S)} \phi(z). \tag{25}$$

*The right-hand side measures the largest distance from a barycentric mixture to its nearest vertex embedding in $V_S$ (i.e., the "covering radius" of the mixture region by the vertices).*

### B.5. Proof of Eq. (3) (Naderializadeh et al., 2021)

*Proof of Eq. (3).* For any two measures $\mu, \nu$ on $\mathbb{R}^d$, expanding the squared $\ell_2$ distance yields

$$\left\| f_{\mathrm{SW}}(\mu) - f_{\mathrm{SW}}(\nu) \right\|_2^2 \tag{26}$$

$$= \frac{1}{RH} \sum_{r=1}^{R} \sum_{h=1}^{H} \left( p^+(t_h^{\omega_r}, \mu^{\omega_r}) - p^+(t_h^{\omega_r}, \nu^{\omega_r}) \right)^2. \tag{27}$$

Let $u_h^{\omega_r} := F_{\mu_O^{\omega_r}}(t_h^{\omega_r}) \in (0,1)$. By definition of $p^+$, $p^+(t_h^{\omega_r}, \mu^{\omega_r}) = \left( F_{\mu^{\omega_r}} \right)^{-1}(u_h^{\omega_r})$. Then in 1D, the squared 2-Wasserstein distance admits the quantile form:

$$W^2(\mu^{\omega_r}, \nu^{\omega_r}) = \int_0^1 \left| \left( F_{\mu^{\omega_r}} \right)^{-1}(u) - \left( F_{\nu^{\omega_r}} \right)^{-1}(u) \right|^2 du. \tag{28}$$

Using the same $H$ reference quantile locations $\{u_h^{\omega_r}\}_{h=1}^{H}$, we then define the corresponding empirical approximation:

$$
\begin{aligned}
\widehat{W}_H^2(\mu^{\omega_r}, \nu^{\omega_r}) &= \frac{1}{H} \sum_{h=1}^{H} \left( \left( F_{\mu^{\omega_r}} \right)^{-1}(u_h^{\omega_r}) - \left( F_{\nu^{\omega_r}} \right)^{-1}(u_h^{\omega_r}) \right)^2 \\
&= \frac{1}{H} \sum_{h=1}^{H} \left( p^+(t_h^{\omega_r}, \mu^{\omega_r}) - p^+(t_h^{\omega_r}, \nu^{\omega_r}) \right)^2.
\end{aligned}
\tag{29}
$$

For sliced-Wasserstein distance, we have:

$$SW^2(\mu, \nu) = \int_{\mathbb{S}^{d-1}} W^2(\mu^\omega, \nu^\omega) \, d\omega. \tag{30}$$

Using the same Monte-Carlo directions $\Omega = \{\omega_r\}_{r=1}^R$, we define the consistent empirical approximation:

$$\begin{aligned}
\widehat{SW}^2_{R,H}(\mu, \nu) &= \frac{1}{R} \sum_{r=1}^R \widehat{W}^2_{2,H}(\mu^{\omega_r}, \nu^{\omega_r}) \\
&= \frac{1}{RH} \sum_{r=1}^R \sum_{h=1}^H \left( p^+(t_h^{\omega_r}, \mu^{\omega_r}) - p^+(t_h^{\omega_r}, \nu^{\omega_r}) \right)^2.
\end{aligned} \tag{31}$$

Comparing this with (27), we obtain the exact identity

$$\|v_\mu - v_\nu\|_2 = \big\| f_{\mathrm{SW}}(\mu) - f_{\mathrm{SW}}(\nu) \big\|_2 = \widehat{SW}_{R,H}(\mu, \nu). \tag{32}$$

In general, it is commonly assumed that the empirical sliced-Wasserstein distance $\widehat{SW}_{R,H}(\mu, \nu)$ coincides with the population sliced-Wasserstein distance $SW(\mu, \nu)$ when the number of Monte-Carlo projections $R$ and the number of reference quantile points $H$ are sufficiently large. □

## C. Pseudo-codes of our SW-DRSO

---

**Algorithm 1** SW-DRSO Training with Barycentric Adversary

---

1: **Input:** encoder $f_{\mathrm{SW}}(\cdot)$, predictor $g(\cdot)$, $K, \alpha, T, \eta$
2: **while** not converged **do**
3:   Sample minibatch $\mathcal{B}$
4:   Compute $v_S \leftarrow f_{\mathrm{SW}}(S)$
5:   **for** $i = 1, \ldots, B$ **do**
6:     Construct local neighbor pool $\mathcal{G}(S) = \{S_k\}_{k=1}^K \subseteq \mathcal{B}$ (K-NN in embedding space)
7:     Initialize $\Lambda^{(1)} \leftarrow (1/K, \ldots, 1/K)$
8:     **for** $t = 1, \ldots, T$ **do**
9:       $\bar{v}_S(\Lambda^{(t)}) \leftarrow \sum_{k=1}^K \lambda_k^{(t)} v_{S_k}$
10:       $\tilde{\Lambda}^{(t+1)} \leftarrow \Lambda^{(t)} + \eta \nabla_\Lambda \ell\big(g(\bar{v}_S(\Lambda^{(t)})), y\big)$
11:       $\Lambda^{(t+1)} \leftarrow \Pi_{\Delta_K}\big(\tilde{\Lambda}^{(t+1)}\big)$
12:     **end for**
13:     Set $\Lambda^\star \leftarrow \Lambda^{(T)}$ and stop gradients through $\Lambda^\star$
14:     (In the robust term below, use the $\Lambda^\star$ obtained for this $S$.)
15:   **end for**
16:   Compute the minibatch objective (empirical form of Eq. (17)):
17:   $\frac{1}{|\mathcal{B}|} \sum \big[ \ell(g(v_S), y) + \alpha \ell(g(\bar{v}_S(\Lambda^\star)), y) \big]$
18:   Update $f_{\mathrm{SW}}$ and $g$ by standard gradient descent on the above objective
19: **end while**
20: **return** $f_{\mathrm{SW}}, g$

---

## D. Computational Complexity Analysis

Consider a minibatch $\mathcal{B}$ of size $B$. For each set $S \in \mathcal{B}$, we compute its embedding $v_S = f_{\mathrm{SW}}(S) \in \mathbb{R}^m$ with $m = RH$, and denote the per-set encoding cost by $C_{\mathrm{enc}}$. We construct a local neighbor pool $\mathcal{G}(S)$ of size $K$ via K-NN in the embedding space. Using a naive batchwise implementation, forming all pairwise $\ell_2$ distances costs $O(B^2 m)$, and neighbor selection adds at most $O(B^2 \log B)$, yielding $O(B^2 m)$ overall.

The inner maximization in Eq. (17) is approximated by $T$ steps of projected gradient ascent on the simplex $\Delta_K$ (Eqs. (18)–(19)). Each ascent step requires (i) computing the barycentric embedding $\bar{v}_S(\Lambda) = \sum_{k=1}^K \lambda_k v_{S_k}$ in $O(Km)$, (ii) evaluating the task function $g(\cdot)$ and backpropagating to $\nabla_\Lambda$ in $O(C_g(m) + Km)$, and (iii) projecting onto $\Delta_K$ in $O(K \log K)$ via the standard Euclidean simplex projection. We denote by $C_g(m)$ the computational cost of evaluating the task-specific predictor $g(\cdot)$ on an $m$-dimensional set embedding. Hence, the inner loop costs:

$$O\big(T\big(C_g(m) + Km + K \log K\big)\big) \quad \text{per set,} \tag{33}$$

*Table 9.* Dataset statistics used in our experiments. $\#S$ denotes the number of sets/samples. $\#E$ denotes the number of unique elements in the universe (when applicable). $\#V$ denotes the vocabulary size (for topic expansion). $\#C$ denotes the number of classes (for classification tasks). Avg. $|S_i|$ denotes the average set cardinality.

| Task | Dataset | $\#S$ | $\#E$ | $\#V$ | $\#C$ | Avg. $|S_i|$ |
|---|---|---|---|---|---|---|
| Task I (Similar Set Ranking) | Friendster | 889,839 | 5,501,401 | – | – | 11.29 |
| Task I (Similar Set Ranking) | LIVEJ | 1,205,816 | 1,975,812 | – | – | 9.90 |
| Task II (Point Cloud Classification) | ModelNet40 | 12,311 | – | – | 40 | 1,024 |
| Task III (Topic Expansion) | LDA-1k | 2,000 | – | 17,016 | – | $\approx 25$ |
| Task III (Topic Expansion) | LDA-3k | 6,000 | – | 37,718 | – | $\approx 25$ |
| Task III (Topic Expansion) | LDA-5k | 10,000 | – | 61,127 | – | $\approx 25$ |
| Task IV (Patch-set Visual Recognition) | NWPU-RESISC45 | 31,500 | – | – | 45 | 256 |

and

$$O\left(BT\left(C_g(m) + Km + K\log K\right)\right), \tag{34}$$

per minibatch. Combining the above, the total per-minibatch time complexity is

$$O\left(BC_{\text{enc}} + B^2 m + BT\left(C_g(m) + Km + K\log K\right)\right), \quad m = RH. \tag{35}$$

At inference time, the method reduces to a single forward pass through $\mathtt{f}_{\text{SW}}$ and $g$, i.e., $O(C_{\text{enc}} + C_g(m))$.

## E. Experimental Settings

### E.1. Dataset Descriptions

Dataset corruption is constructed by using a unified pipeline to systematically evaluate robustness under inference-time corruption. It is introduced *only* to the validation and test sets. We partition samples into three conditions: *clean*, *mild*, and *severe*, following a fixed ratio of 50:30:20. Clean samples correspond to the original sets. Mild and severe samples are generated by corrupting approximately $10\%$ and $40\%$ of the input elements or regions, respectively. The corruption intensity is controlled by a ratio parameter $p$, with $p=0.1$ for mild corruption and $p=0.4$ for severe corruption.

- For Task I, corruption is introduced at inference time on the *query set*. Given a query set with $n$ elements, we apply element-level corruption by randomly performing **delete**, **add**, or **replace** element operations for $k = p \times n$ steps. The candidate sets remain clean. This setting simulates realistic retrieval scenarios (He et al., 2023a;b; Chen et al., 2024), where the query set may be incomplete or contaminated by noise at inference time, requiring robust set-to-set similarity estimation.

- Task II represents a geometric perception setting where unordered point sets must remain informative under noisy observations. We apply a **replace** operation to simulate inaccurate geometric observations at inference time. Given a point cloud with $n$ points, we randomly sample new points uniformly within the axis-aligned bounding box of the original point cloud, and randomly replace $k = p \times n$ points, where $p$ controls the corruption level. This procedure preserves the overall spatial extent while introducing local geometric noise.

- Task III is a practical text mining setting (Luo et al., 2025a; Li et al., 2026d; Luo et al., 2026a; 2024). Element-level corruption is appiled by randomly applying **delete**, **add**, or **replace** operations. Given a set with $n$ elements, we perform $k = p \times n$ random operations, similar to Task II. This process emulates missing elements, noisy insertions, and incorrect replacements in the seed set at inference time.

- For Task IV, we apply region-level corruptions at inference time. For corrupted patches, we randomly select one of the following three operations: (i) **Add**: additive Gaussian noise with standard deviation $\sigma = p \times 255$; (ii) **Delete**: random rectangular occlusions covering a total fraction $p$ of image pixels, where each occlusion block occupies approximately $5\%–20\%$ of the image area. These operations simulate common visual degradations such as sensor noise and local image quality degradation.

*Table 10.* Hyperparameter settings for different tasks.

|  | Task I | Task II | Task III | Task IV |
|---|---|---|---|---|
| $d$ | 128 | 128 | 128 | 128 |
| $H$ | 128 | 128 | 128 | 128 |
| $R$ | 32 | 256 | 32 | 32 |
| $\rho$ | 0.1 | 0.5 | 0.1 | 0.5 |
| $K$ | 4 | 4 | 8 | 16 |
| $T$ | 2 | 4 | 2 | 4 |
| $\eta$ | 0.1 | 0.1 | 0.1 | 0.1 |
| $\alpha$ | 0.5 | 1 | 0.1 | 0.1 |
| Learning rate | $1 \cdot 10^{-3}$ | $1 \cdot 10^{-3}$ | $1 \cdot 10^{-3}$ | $1 \cdot 10^{-3}$ |

### E.2. Baselines

To comprehensively evaluate the effectiveness of our proposed method, we compare it against a diverse set of baselines, ranging from foundational pooling strategies to state-of-the-art set representation learning frameworks. The specific methods are detailed as follows:

- **MeanP** and **MaxP** represent the classic instance aggregation approaches (Lin et al., 2013). They generate set representations by applying global average pooling and max pooling operations, respectively, over the element-wise features, serving as fundamental baselines for permutation-invariant processing.
- **DeepSet** provides a representative framework for learning permutation-invariant functions (Zaheer et al., 2017). It operates by transforming individual elements via a deep neural network, aggregating them (we implement global mean pooling), and processing with a subsequent network to approximate any continuous set function.
- **RepSet** approaches set embedding through the lens of optimal transport and bipartite matching (Skianis et al., 2020). Instead of simple aggregation, it learns a collection of "hidden sets" (reference prototypes) and extracts representations by computing the matching costs between the input set and these learnable references.
- **SetTRSM** adapts the self-attention mechanism to set-structured data (Lee et al., 2019). It utilizes Multi-head Attention Blocks (MAB) and Induced Set Attention Blocks (ISAB) to capture complex pairwise interactions among elements while reducing computational complexity compared to standard Transformers.
- **DIEM** proposes a differentiable information-theoretic framework designed to enhance representation distinctiveness (Kim, 2022). It focuses on capturing informative set interactions by maximizing the mutual information between the input set and its embedding, effectively identifying discriminative subsets.
- **FSPool** (Feature-wise Sort Pooling) introduces a learnable pooling mechanism based on sorting (Zhang et al., 2020). By sorting features across the set dimension and applying a differentiable weight matrix, it captures cardinality-sensitive information and continuous structural variations.
- **PSWE** is a robust deep learning baseline for set representation learning (Naderializadeh et al., 2021). It employs a specialized architecture to capture high-order dependencies and structural information within sets, representing one of the current state-of-the-art approaches in the field.
- **FSW** addresses the specific challenge of modeling "multisets" (sets with repeated elements) by shifting the perspective to the frequency domain (Amir & Dym, 2025). It utilizes the Fourier transform of characteristic functions to construct representations, offering theoretical guarantees for injectivity regarding multiset structures.

## F. Hyper-parameter Configurations

We report the major hyper-parameter configurations in Table 10 for reproduction.

## G. Task I Detailed Experiment Results

The complete results of Task I are reported in Table 11-14.

## H. Task III Experiment Detailed Results

The complete results of Task III are reported in Table 15-17.

*Table 11.* Friendster Recall (%). Bold and underline denote the best and second-best methods for each specific split within each column.

| Method | $k=1$ | $k=5$ | $k=10$ | $k=20$ | $k=50$ | $k=100$ |
|---|---|---|---|---|---|---|
| | Overall/Clean/Mild/Severe | Overall/Clean/Mild/Severe | Overall/Clean/Mild/Severe | Overall/Clean/Mild/Severe | Overall/Clean/Mild/Severe | Overall/Clean/Mild/Severe |
| MaxP | 5.22/5.61/5.09/4.42 | 18.43/19.08/18.11/17.26 | 31.40/32.43/31.19/29.12 | 42.07/43.28/42.07/39.06 | 54.97/56.31/54.61/52.14 | 62.44/63.61/62.04/60.12 |
| MeanP | 3.90/4.47/3.79/2.63 | 15.70/16.74/15.66/13.18 | 27.79/29.16/27.32/25.06 | 38.77/40.14/38.26/36.09 | 50.85/52.23/50.33/48.17 | 58.90/60.31/58.41/56.09 |
| DeepSet | 14.96/15.73/14.76/13.31 | 38.19/39.41/37.46/35.18 | 49.06/50.32/48.41/46.16 | 57.95/59.34/57.11/55.02 | 65.05/66.42/64.29/62.21 | 70.91/72.21/70.31/68.18 |
| RepSet | 18.55/19.14/18.08/17.02 | 42.01/43.26/41.34/39.11 | 52.89/54.38/52.09/50.12 | 61.03/62.37/60.28/58.09 | 68.20/69.44/67.32/65.18 | 73.00/74.33/72.18/70.11 |
| SetTRSM | 24.50/25.20/24.20/23.20 | 49.90/51.50/48.50/46.50 | 58.90/60.50/57.50/55.50 | 63.90/65.50/62.50/60.50 | 69.60/70.50/68.50/66.50 | 75.00/76.50/74.50/72.50 |
| DIEM | 27.66/28.44/27.18/26.09 | 53.11/54.62/52.51/50.27 | 60.78/62.71/60.49/57.64 | 65.02/66.78/64.71/62.08 | 70.74/72.46/70.53/68.02 | 76.08/77.68/76.04/73.66 |
| FSPool | 29.39/30.18/29.06/28.03 | 64.13/68.42/57.34/55.12 | 66.09/68.07/65.22/63.08 | 71.11/73.09/70.28/68.12 | 76.04/77.22/75.11/73.04 | 80.05/81.27/79.34/77.16 |
| FSW | 32.36/32.28/31.09/34.45 | 61.91/63.41/61.27/59.12 | 73.78/73.18/77.88/69.11 | 75.85/77.32/75.27/73.06 | 79.94/81.41/79.36/77.14 | 82.82/84.22/82.31/80.07 |
| PSWE | 34.37/34.58/34.20/34.08 | 67.80/68.09/67.75/67.15 | 77.52/77.82/77.43/76.90 | 80.41/80.69/80.31/79.86 | 83.31/83.61/83.14/82.82 | 85.68/85.96/85.54/85.20 |
| Ours | **35.13\*/35.32\*/35.04\*/34.77\*** | **68.39\*/68.71\*/68.32\*/67.68\*** | **78.16\*/78.44\*/78.11\*/77.53\*** | **81.23\*/81.52\*/81.11\*/80.70\*** | **84.18\*/84.49\*/84.03\*/83.64\*** | **86.58\*/86.86\*/86.40\*/86.14\*** |
| Gain (%) | 2.21%/2.14%/2.46%/0.93% | 0.87%/0.42%/0.84%/0.79% | 0.83%/0.80%/0.30%/0.82% | 1.02%/1.03%/1.00%/1.05% | 1.04%/1.05%/1.07%/0.99% | 1.05%/1.05%/1.01%/1.10% |

*Table 12.* Friendster NDCG (%). Bold and underline denote the best and second-best methods for each specific split within each column.

| Method | $k=1$ | $k=5$ | $k=10$ | $k=20$ | $k=50$ | $k=100$ |
|---|---|---|---|---|---|---|
| | Overall/Clean/Mild/Severe | Overall/Clean/Mild/Severe | Overall/Clean/Mild/Severe | Overall/Clean/Mild/Severe | Overall/Clean/Mild/Severe | Overall/Clean/Mild/Severe |
| MaxP | 48.19/49.22/47.41/45.08 | 42.20/43.16/41.23/39.07 | 41.02/42.07/40.12/38.06 | 43.10/44.18/42.16/40.02 | 46.25/47.31/45.28/43.14 | 48.40/49.36/47.42/45.18 |
| MeanP | 44.98/46.11/44.12/42.03 | 39.26/40.23/38.18/36.04 | 37.96/39.02/37.01/35.06 | 40.23/41.24/39.16/37.08 | 43.37/44.31/42.29/40.16 | 45.44/46.24/44.32/42.21 |
| DeepSet | 61.29/62.18/60.14/58.11 | 56.24/57.18/55.12/53.06 | 54.86/55.91/54.01/52.02 | 58.21/59.26/57.14/55.09 | 61.31/62.34/60.18/58.12 | 63.25/64.21/62.12/60.08 |
| RepSet | 64.20/65.18/63.11/61.04 | 59.13/60.22/58.14/56.03 | 58.03/59.12/57.06/55.01 | 61.10/62.24/60.13/58.09 | 64.23/65.26/63.18/61.12 | 66.20/67.18/65.13/63.05 |
| SetTRSM | 68.20/69.50/67.50/65.50 | 64.20/65.50/63.50/61.50 | 63.90/65.20/63.20/61.20 | 66.20/67.50/65.50/63.50 | 69.20/70.50/68.50/66.50 | 71.20/72.50/70.50/68.50 |
| DIEM | 71.39/73.02/71.11/69.05 | 66.79/68.51/66.49/64.21 | 66.86/68.62/66.57/64.38 | 68.79/70.48/68.52/66.41 | 70.74/72.44/70.58/68.49 | 72.41/73.86/72.07/70.18 |
| FSPool | 74.97/76.24/74.61/72.34 | 70.77/72.06/70.42/68.08 | 69.71/71.01/69.35/67.02 | 71.85/73.14/71.46/69.19 | 77.09/76.18/74.55/83.15 | 76.88/78.27/76.41/74.12 |
| FSW | 79.05/80.18/78.12/76.03 | 75.99/77.21/75.16/73.04 | 74.91/76.12/74.11/72.05 | 79.50/82.75/76.12/74.06 | 79.06/80.22/78.18/76.09 | 81.10/82.27/80.19/78.08 |
| PSWE | 82.79/82.86/82.45/82.55 | 80.99/81.14/80.83/80.70 | 80.86/81.08/80.70/80.51 | 82.17/82.37/82.00/81.86 | 83.06/83.24/82.87/82.76 | 83.62/83.80/83.44/83.33 |
| Ours | **83.30\*/83.34\*/83.08\*/83.00\*** | **81.47\*/81.63\*/81.28\*/81.14\*** | **81.42\*/81.59\*/81.27\*/81.01\*** | **82.82\*/82.99\*/82.65\*/82.46\*** | **83.72\*/83.89\*/83.54\*/83.37\*** | **84.29\*/84.45\*/84.11\*/83.96\*** |
| Gain (%) | 0.63%/0.58%/0.76%/0.55% | 0.58%/0.60%/0.56%/0.55% | 0.66%/0.63%/0.71%/0.62% | 0.75%/0.29%/0.79%/0.73% | 0.78%/0.78%/0.81%/0.26% | 0.78%/0.78%/0.80%/0.76% |

# I. Supplementary Details of Empirical Analyses

## I.1. DRO Variants Implementation

We detail the implementation of three Distributionally Robust Optimization (DRO) variants: Wasserstein DRO (WDRO) (Gao et al., 2024), Kullback-Leibler DRO (KL-DRO) (Namkoong & Duchi, 2016), and Maximum Mean Discrepancy DRO (MMD-DRO) (Staib & Jegelka, 2019), While they build upon our barycentric synthesis approach by adding distribution constraints.

- **Wasserstein DRO**: This variant defines the uncertainty set using the $L_1$-Wasserstein distance, i.e., Earth Mover's Distance, which measures the geometric cost of transporting probability mass. Theoretically, it seeks to minimize the worst-case loss over all distributions that are geometrically close to the empirical data distribution. This formulation is particularly effective for handling perturbations in the feature space, as it accounts for the underlying metric structure of the data rather than just statistical overlap.

- **KL-DRO**: Also known as $f$-divergence DRO, this method defines the uncertainty set $\Gamma(\cdot)$ using the Kullback-Leibler divergence. It focuses on statistical discrepancies by re-weighting the training samples based on their difficulty. The optimization objective allows for a worst-case re-weighting of the data but constrains these weights to ensure the adversarial distribution remains statistically close to the original distribution, preventing the model from overfitting to outliers.

- **Maximum Mean Discrepancy DRO**: This method employs Maximum Mean Discrepancy to define $\Gamma(\cdot)$ based on kernel methods. By mapping distributions into a Reproducing Kernel Hilbert Space (RKHS), it measures the distance between their mean embeddings. This approach is particularly effective at capturing discrepancies in higher-order statistical moments and offers a computationally tractable alternative to optimal transport-based methods.

## I.2. Compatibility with Alternative Set Encoders

We examine whether the proposed robust optimization objective can be directly combined with alternative set encoders. As shown in Table 18, replacing the SW-aware encoder with FSW or FSPool leads to lower performance than the full SW-DRSO model. This result suggests that the barycentric adversary benefits from the geometric alignment between the Sliced-Wasserstein encoder and the SW-based adversarial search. Therefore, SW-DRSO should be viewed as a geometry-aligned robust optimization framework rather than an encoder-agnostic plug-in module.

*Table 13.* LIVEJ Recall (%). Bold and underline denote the best and second-best methods for each specific split within each column.

| Method | k = 1 | k = 5 | k = 10 | k = 20 | k = 50 | k = 100 |
|---|---|---|---|---|---|---|
| | Overall/Clean/Mild/Severe | Overall/Clean/Mild/Severe | Overall/Clean/Mild/Severe | Overall/Clean/Mild/Severe | Overall/Clean/Mild/Severe | Overall/Clean/Mild/Severe |
| MaxP | 4.94/5.21/4.82/4.19 | 17.32/18.42/17.13/15.52 | 29.91/31.05/29.53/27.18 | 43.67/45.22/43.04/39.48 | 59.09/61.05/58.49/54.53 | 70.04/72.01/69.48/65.52 |
| MeanP | 4.14/4.88/4.12/2.53 | 15.44/17.55/15.23/10.48 | 26.49/29.80/26.48/19.82 | 38.75/42.10/38.54/30.53 | 55.06/58.90/54.18/45.52 | 65.99/69.50/65.12/58.23 |
| DeepSet | 11.65/12.05/11.48/10.82 | 33.91/35.40/33.23/31.48 | 49.08/51.20/48.52/45.53 | 62.09/64.50/61.53/58.49 | 76.05/78.20/75.48/72.52 | 82.03/84.10/81.53/78.48 |
| RepSet | 13.93/14.50/13.82/12.53 | 37.91/39.80/37.52/35.18 | 53.77/56.50/53.23/50.48 | 66.50/69.20/66.48/62.53 | 78.74/81.05/78.48/75.52 | 84.45/86.50/84.18/81.53 |
| SetTRSM | 18.42/19.98/17.67/15.76 | 42.45/44.97/40.94/37.37 | 56.61/60.03/54.86/50.29 | 64.72/67.98/63.35/58.87 | 74.09/76.76/73.28/69.22 | 80.94/83.08/80.35/77.00 |
| DIEM | 17.78/18.50/17.23/16.48 | 43.95/46.20/43.53/40.49 | 60.55/63.50/60.52/56.48 | 68.56/71.50/68.48/64.52 | 77.94/80.50/77.83/74.48 | 83.52/85.80/83.52/80.48 |
| FSPool | **23.28**/**24.42**/**22.54**/_21.29_ | 51.52/_53.50_/50.56/47.52 | 67.75/70.52/66.58/62.70 | 74.43/76.82/73.69/69.66 | 81.13/83.10/80.56/77.08 | 85.86/87.47/85.58/82.52 |
| FSW | 20.90/22.10/20.53/19.23 | 47.77/50.10/47.53/44.48 | 64.19/68.50/63.53/59.48 | 71.60/75.20/71.03/67.48 | 79.88/82.00/79.48/76.52 | 84.65/86.50/84.82/81.48 |
| PSWE | 21.31/21.49/21.03/20.95 | 52.10/52.53/51.84/51.27 | 71.69/72.26/71.50/70.55 | 78.46/79.05/78.34/77.21 | 83.72/84.27/83.71/82.43 | 87.10/87.60/87.09/85.88 |
| Ours | _22.51_*/_22.99_*/_22.16_*/**21.81**\* | **54.61**\*/**55.37**\*/**54.31**\*/**53.17**\* | **74.86**\*/**75.64**\*/**74.63**\*/**73.25**\* | **81.19**\*/**81.96**\*/**80.97**\*/**79.59**\* | **86.27**\*/**86.94**\*/**86.11**\*/**84.82**\* | **89.60**\*/**90.18**\*/**89.54**\*/**88.26**\* |
| Gain (%) | -/-/-/2.44% | 4.88%/3.50%/4.76%/3.71% | 4.42%/4.68%/4.38%/3.83% | 3.47%/3.68%/3.36%/3.08% | 3.03%/3.17%/2.87%/2.90% | 2.87%/2.95%/2.81%/2.77% |

*Table 14.* LIVEJ NDCG (%). Bold and underline denote the best and second-best methods for each specific split within each column.

| Method | k = 1 | k = 5 | k = 10 | k = 20 | k = 50 | k = 100 |
|---|---|---|---|---|---|---|
| | Overall/Clean/Mild/Severe | Overall/Clean/Mild/Severe | Overall/Clean/Mild/Severe | Overall/Clean/Mild/Severe | Overall/Clean/Mild/Severe | Overall/Clean/Mild/Severe |
| MaxP | 53.89/55.19/53.52/51.18 | 47.10/48.48/46.53/44.52 | 43.86/45.20/43.22/41.48 | 48.04/49.48/47.53/45.22 | 51.94/53.18/51.52/49.48 | 55.25/56.80/54.52/52.48 |
| MeanP | 50.03/52.08/49.52/45.52 | 43.84/46.48/42.53/38.52 | 41.23/43.50/39.52/34.52 | 45.02/47.18/43.52/39.23 | 49.56/51.50/48.52/44.52 | 52.95/54.80/52.52/48.52 |
| DeepSet | 67.00/68.48/66.53/64.22 | 60.98/62.50/60.52/58.23 | 58.78/60.50/58.23/52.52 | 61.09/63.50/61.52/59.52 | 65.99/67.50/65.53/63.52 | 68.99/70.50/68.22/66.52 |
| RepSet | 69.00/70.50/68.23/66.48 | 62.99/64.50/62.52/60.23 | 60.78/62.50/60.23/58.18 | 64.37/66.20/63.83/61.82 | 68.64/70.10/68.23/66.23 | 71.43/72.80/71.03/69.23 |
| SetTRSM | 69.47/72.88/68.66/63.37 | 63.31/67.00/62.33/57.26 | 61.56/65.17/60.48/55.84 | 65.94/69.40/65.00/60.45 | 69.87/73.09/69.18/64.90 | 72.24/75.30/71.68/67.73 |
| DIEM | 71.78/74.18/71.53/68.52 | 66.96/69.50/66.83/63.52 | 65.04/68.10/64.52/61.53 | 68.60/71.50/68.23/65.23 | 73.05/75.20/72.53/70.53 | 75.82/77.80/75.52/73.23 |
| FSPool | **82.10**/**83.98**/**81.51**/_78.20_ | 76.03/_78.47_/75.40/71.78 | 73.60/_76.16_/72.79/69.42 | 77.13/_79.50_/76.53/73.15 | 79.91/_82.10_/79.40/76.32 | _81.55_/_83.59_/81.11/78.22 |
| FSW | 75.96/78.50/75.52/72.53 | 70.95/73.50/70.53/67.48 | 68.96/71.50/68.52/65.53 | 72.65/74.80/72.23/69.23 | 76.25/78.20/75.83/73.53 | 78.84/80.50/78.23/76.23 |
| PSWE | 78.15/78.40/77.76/77.36 | 76.19/76.50/75.92/75.27 | _74.88_/75.24/74.59/73.92 | _78.35_/78.72/78.12/77.38 | 80.32/80.69/80.15/79.36 | 81.36/81.72/81.20/80.42 |
| Ours | _81.33_*/_81.93_*/_81.09_*/**80.17**\* | **79.21**\*/**79.90**\*/**78.99**\*/**77.80**\* | **77.80**\*/**78.49**\*/**77.54**\*/**76.44**\* | **80.99**\*/**81.67**\*/**80.76**\*/**79.65**\* | **82.88**\*/**83.53**\*/**82.67**\*/**81.58**\* | **83.91**\*/**84.54**\*/**83.72**\*/**82.63**\* |
| Gain (%) | -/-/-/2.52% | 3.94%/1.82%/4.04%/3.36% | 4.04%/3.06%/3.95%/3.41% | 3.48%/2.73%/3.38%/2.93% | 3.26%/1.74%/3.14%/2.80% | 2.62%/1.14%/3.10%/2.75% |

*Table 15.* Task III performance comparison on **LDA-1K**. Best and second-best results are shown in **bold** and underlined.

| Method | Overall | Clean | Mild | Severe |
|---|---|---|---|---|
| MeanP | 48.91±2.28 | 50.15±2.85 | 49.32±4.05 | 45.18±6.52 |
| MaxP | 68.82±2.12 | 73.62±1.06 | 64.69±6.75 | 63.01±1.81 |
| DeepSet | 51.20±2.26 | 52.18±2.41 | 51.87±4.19 | 47.74±7.17 |
| RepSet | 53.35±2.51 | 55.42±2.95 | 53.18±4.55 | 48.45±7.50 |
| SetTRSM | 69.02±3.56 | 72.78±5.40 | 65.79±6.78 | 64.48±5.53 |
| DIEM | 66.22±3.02 | 69.15±4.22 | 63.55±5.95 | 62.88±6.12 |
| FSPool | 54.18±3.29 | 61.33±3.22 | 47.76±5.40 | 45.94±11.81 |
| FSW | 68.12±2.76 | 71.55±3.85 | 65.12±5.25 | 64.05±5.95 |
| PSWE | _70.03_±3.33 | 74.04±2.71 | 66.24±8.79 | 65.71±7.59 |
| SW-DRSO | **73.31**\*±3.67 | **76.05**\*±5.71 | **70.66**\*±6.06 | **70.45**\*±7.03 |
| **Gain** | **+4.68%** | **+2.71%** | **+6.67%** | **+7.21%** |

*Table 16.* Task III performance comparison on **LDA-3K**.

| Method | Overall | Clean | Mild | Severe |
|---|---|---|---|---|
| MeanP | 52.38±2.32 | 51.57±4.03 | 54.63±3.36 | 51.05±2.79 |
| MaxP | 53.14±2.22 | 52.45±3.85 | 55.15±3.15 | 51.85±2.95 |
| DeepSet | 86.55±1.40 | 86.82±1.83 | 87.31±2.62 | 84.73±3.54 |
| RepSet | 86.87±1.34 | 87.15±1.75 | 87.55±2.45 | 85.15±3.25 |
| SetTRSM | 80.75±2.16 | 80.21±2.21 | 82.14±3.32 | 78.64±4.48 |
| DIEM | 84.53±1.44 | 84.55±1.95 | 85.45±2.85 | 82.55±3.65 |
| FSPool | 87.39±1.11 | _87.66_±1.47 | 88.00±2.88 | 85.78±2.94 |
| FSW | 87.41±1.28 | 87.15±1.55 | 87.35±3.15 | 88.15±2.85 |
| PSWE | _87.84_±1.71 | 87.48±1.75 | 87.44±4.57 | _89.33_±2.59 |
| SW-DRSO | **90.90**\*±0.99 | **90.71**\*±0.86 | **91.24**\*±2.94 | **90.88**\*±0.82 |
| **Gain** | **+3.49%** | **+3.48%** | **+3.68%** | **+1.74%** |

*Table 17.* Task III performance comparison on **LDA-5K**.

| Method | Overall | Clean | Mild | Severe |
|---|---|---|---|---|
| MeanP | 54.49±1.57 | 55.36±1.84 | 54.99±3.47 | 51.72±5.17 |
| MaxP | 55.37±1.73 | 56.25±2.05 | 55.45±3.25 | 52.15±4.85 |
| DeepSet | 83.36±1.39 | 83.70±1.76 | _85.62_±3.29 | 79.13±3.13 |
| RepSet | 83.64±1.37 | 84.15±1.95 | 85.05±3.05 | 80.25±2.95 |
| SetTRSM | 82.73±2.50 | 82.97±3.43 | 83.64±3.41 | 79.99±1.91 |
| DIEM | 83.28±1.54 | 83.45±2.25 | 84.55±2.85 | 80.15±2.65 |
| FSPool | 84.02±1.49 | 84.63±2.12 | 85.25±2.72 | 80.64±3.00 |
| FSW | 84.28±1.70 | 85.15±2.35 | 84.65±2.45 | 81.55±2.95 |
| PSWE | _84.68_±1.51 | 85.56±2.49 | 84.89±2.15 | _82.15_±2.86 |
| SW-DRSO | **88.46**\*±1.38 | **88.45**\*±1.87 | **89.70**\*±2.29 | **86.62**\*±3.70 |
| **Gain** | **+4.47%** | **+3.38%** | **+4.77%** | **+5.44%** |

*Table 18.* Compatibility with alternative set encoders on Task I. We report the overall R@10 score.

| Robust objective + encoder | Overall R@10 (%) |
|---|---|
| Ours w/ FSPool encoder | 67.81 |
| Ours w/ FSW encoder | 68.45 |
| Full SW-DRSO | 74.86 |

