# OpenReview forum: "Distributionally Robust Set Representation Learning Under Inference-Time Element Corruption"
_ICML.cc/2026/Conference — ICML 2026 regular_

### Official Review · Reviewer_XAGg · 2026-03-12

**Soundness:** 3
**Presentation:** 2
**Significance:** 3
**Originality:** 2
**Overall Recommendation:** 4
**Confidence:** 3

**Summary:**

The paper studies how to make set representation learning models robust to element-level corruption at inference time, where some elements in a set may be missing or perturbed. To address this, the authors formulate the problem as a Sliced-Wasserstein distributionally robust optimization (SW-DRO) task and approximate the adversarial search by synthesizing barycentric perturbations in the latent space of a Sliced-Wasserstein set encoder.

**Compliance With Llm Reviewing Policy:**

Affirmed.

**Final Justification:**

The authors' response has addressed my queries. Although some inherent limitations of the framework remain, I believe the paper is technically solid and suitable for publication, so I maintain my "Weak Accept" recommendation.

**Key Questions For Authors:**

### Questions

1. Consistency with Element-Level Corruption. The paper motivates the problem as element-level corruption, where only a subset of elements in a set is perturbed. However, the proposed method perturbs the set distribution in the representation space. It is therefore unclear whether this formulation faithfully captures the intended element-level corruption scenario.

**Limitations:**

No.

**Strengths And Weaknesses:**

### Strengthes

1. Geometry-consistent adversarial formulation. The paper exploits the structure of the Sliced-Wasserstein encoder to show that convex combinations of set embeddings correspond to Wasserstein barycenters of the underlying set distributions. This enables the adversarial search to be parameterized by simplex weights over sets, turning an intractable combinatorial search over corrupted sets into a low-dimensional differentiable optimization while preserving a valid distributional interpretation.
2. Comprehensive experimental evaluation. The method is validated on four downstream tasks spanning multiple data modalities, including social networks, point clouds, text topics, and visual patch sets, and is compared against a wide range of representative SRL baselines such as pooling-based, attention-based, and optimal-transport-based methods, which together provide a reasonably thorough demonstration of the proposed approach’s generality and robustness.

### Weaknesses

1. Overly permissive SW-based uncertainty set. The optimization is defined over sets $S'$ within an SW-ball around the empirical distribution $\mu_S$, but it remains unclear whether $S'$  is restricted to empirical distributions with $n$ samples or allowed to represent arbitrary (possibly continuous) probability measures. Since the Sliced-Wasserstein ball is defined over the space of probability distributions, the theoretical formulation permits much richer distributions than finite empirical sets.
2. Dependence on a specific encoder design.The proposed tractable optimization relies on performing perturbations in the latent space of a particular Sliced-Wasserstein set encoder. This dependence is non-trivial, as the barycentric synthesis used in the adversarial search requires the encoder to preserve SW geometry—i.e., convex combinations of embeddings must correspond to Wasserstein barycenters of the underlying set distributions—which does not generally hold for arbitrary set encoders.

---

> ### Author Rebuttal · Authors · 2026-03-31
>
> We are grateful to Reviewer XAGg for the constructive feedback, which helped us better clarify and improve our work. We now provide responses below to address all concerns accordingly.
>
> ---
>
> W1:
>
> Thank you Reviewer XAGg for this insightful feedback. As you commented, once formulated via an SW ball, the uncertainty set naturally lives in the broader space of probability measures and is therefore richer than the space of finite empirical sets. But we respectfully hope to clarify that, `this relaxation is acutally intentional`: if we were to require $S'$ to remain an explicit finite set and perform exhaustive search directly in that space, the inner maximization would be highly intractable. So our SW-based barycentric adversary is therefore introduced as a tractable relaxation to the optmization objective.
>
> ---
>
>
> W2:
>
> Thanks Reviewer XAGg for this important observation. We agree that our optimization is not encoder-agnostic: the barycentric synthesis relies on the embedding space preserving SW geometry, so that convex combinations in latent space correspond to Fréchet means of the underlying set distributions. This is also an intentional design, mainly to achieve `geometric alignment between the encoder and the adversarial search procedure`. We agree that exploring such corruption-aware robust optimization to more general set encoders is a more promising direction, and we will discuss this as future work. Thanks for the suggestion.
>
>
> ---
>
>
> Q1:
>
> We appreciate Reviewer XAGg for this insightful comment. Our main motivation for adopting a distributional formulation is that, the effect of element-level corruption on a set model does not only result in the local modification of one or a few elements, but also the resulting change in the empirical distribution of the set, which may in turn distort the aggregated representation. As the reviewer rightly points out, perturbations in latent space may inevitably introduce effects beyond strict element-level corruption. To control this, we impose an SW-distance constraint so that the perturbed distribution does not deviate too far from the original one. This can be viewed as a soft restriction intended to implicitly capture scenarios in which only part of the set is corrupted, while the overall semantics still remain largely preserved. We will clarify this motivation more explicitly in the method section.
>
> ---
>
>
> *We hope our responses effectively address Reviewer XAGg's concerns. We remain available for any further clarifications and discussions.*

---

> > ### Author Rebuttal · Reviewer_XAGg · 2026-04-03
> >
> > The authors' response has addressed my queries. Although some inherent limitations of the framework remain, I believe the paper is technically solid and suitable for publication, so I maintain my "Weak Accept" recommendation.

---

> > > ### Author Response · Authors · 2026-04-03
> > >
> > > Dear Reviewer XAGg,
> > >
> > > We sincerely thank you for your effort in paper review and for recognizing our work. We remain available to address any further questions or suggestions.

---

### Official Review · Reviewer_WEba · 2026-03-13

**Soundness:** 3
**Presentation:** 3
**Significance:** 3
**Originality:** 3
**Overall Recommendation:** 4
**Confidence:** 4

**Summary:**

Set based models trained on clean sets often break when a few elements are corrupted at test time, such as missing items, outliers, or noisy patches. The paper proposes SW-DRSO, a distributionally robust training method that represents sets as empirical measures, defines plausible corruptions with the Sliced-Wasserstein distance, and uses a barycentric adversary to efficiently approximate worst-case corrupted sets during training. Across four tasksthe method is reported to improve robustness under corruption while preserving performance on clean data.

**Compliance With Llm Reviewing Policy:**

Affirmed.

**Key Questions For Authors:**

- Do the surrogates really take care of difficult corruptions besides interpolations in local neighbourhoods?

- Does the corruption methods used in the experiments actually mimic real world situations? While I very much like the idea of building set functions robust to set element corruption, the problems seems a bit artificial since no real world example of such cases are actually tackled in the experimental section.

- How scalable is the proposed method especially for large sets?

- Can you provide some parameter counts and training curves, corrected for compute, with the baselines given the complexity of the proposed method? My main concern is weather the gain is just due to longer training, larger architectures etc.

- Any sensitivity analysis on the parameters in line 1 of Algorithim 1?

**Limitations:**

Yes

**Strengths And Weaknesses:**

**Strengths**

   - The problem is well motivated and tackles an often unexplored sub problem in the set literature, by placing a focus on inference time robustness to set element corruption.

   - The combinatorial search for worst case corrupted sets in efficiently approximated.

   - The proposed method is well evaluated and the results show improved robustness of the proposed model under element corruption.

   - Extensive ablation study is used to justify design choices.

**Weaknesses**

- It is unclear to me if optimizing the surrogates actually approximates the original worst-case objective in practice.

- The adversaries are built from convex combinations of nearby training embeddings and may not necessarily match corruptions outside the convex hull.

- Corruptions are manually constructed (the main motivation is on real world scenarios of corrupted set data).

 - Parameter counts are not provided (for instance compared to deepsets or set transformer).

  - Optimization of the proposed method is more involved that traditional set-based methods (adversarial optimization, neighbour search etc.) these are not accounted for in the baselines.

- Some of the gains are small (ie. not consisntenly outperforming the baselines as claimed).

---

> ### Author Rebuttal · Authors · 2026-03-31
>
> We sincerely appreciate Reviewer WEba for your professional and detailed feedback. We have addressed each of your comments and hope to clarify these points satisfactorily.
>
> ---
>
> W1:
> Thank you for this important comment.
> - The original worst-case objective requires searching over a combinatorial space of corrupted sets, so exact optimization is computationally intractable. For the same reason, rigorously quantifying the gap between it and our surrogate is also difficult.
> - Then instead of an exhaustive search, our barycentric adversary achieves continuous optimization over convex mixing weights in embedding space. Prop. 4.3 shows it is better, as selecting one discrete neighbor is only a special case of our simplex-constrained barycentric optimization.
> - Empirically, Sec. 5.3.2/Table 4 shows better peformance than the discrete adversary baseline. We also added a random combinational sampling (RCS) experiment on Task 1 (LiveJ) by exploring candidates in batch training below.
> They are more costly with limited gains. We will discuss these additional results in revision.
>
> | R@10(%)/Time(min)| n=2 | n=5 | n=10|
> |---|---|---|---|
> | RCS | 69.45/21.7 | 70.56/23.5 | 71.13/27.4 |
> | RCS with SW-distance constraint | 69.52/22.6 | 71.81/24.8 | 71.74/28.1|
>
> ---
>
> W2 Q1:
> We respectfully hope to clarify that
> - Our adversary is primarily designed as a training-time adversarial opponent that exposes possible optimization directions, rather than as a generator of real-world corruption patterns, which is actually hard to achieve.
> - Then we intentionally restrict adversarial search to the local convex hull within local region. In this sense, the goal is to provide a trainable robust optimization framework for element corruption, not to exhaustively model all out-of-hull corruptions. We will clarify it more explicitly to avoid any ambiguity in revision. Thx!
>
> ---
>
> W3 Q2:
> Thank you for recognizing our work and critical comments. Corruptions in our experiments are manually constructed and grouped into clean/mild/severe settings, mainly for `controlled robustness evaluation under realistically motivated corruption patterns`; they are not arbitrary noise but are designed to reflect task-specific corruption modes. We fully agree that collecting and evaluating on more realistic deployment corruption data is an important next step, and we will state this more clearly as future work.
>
> ---
>
> W4 Q4:
> Thank you Reviewer WEba for constructive suggestions.
> - Using Task 1 as an example, besides the set element embedding table that is shared by all models, the parameters of all baselines and ours are 1.1M/1.1M/0.3M/0.6M/4.3M/2.1M/0.2M/2.5M/1.2M and 1.2M.
> - For our method, adversarial optimization introduces `~1.4M extra training-time parameters`. This can be adjusted in practice, depending on setups such as GPU memory and batch size.
> - But honestly, we do believe that the design of adversarial training, which includes additional intermediate parameters, ultimately improves the model's performance.
>
> Due to rebuttal restrictions, we will add a complete parameter analysis and training curves adjusted by parameter numbers in revision.
>
> ---
>
> W5:
> - The main reason is that prior set-based methods were not designed to address element corruption issue, and therefore do not include corresponding adversarial optimization or robust training mechanisms.
> - We agree that it is valuable to test compatibility of our optimization method with other encoders. Here, we experimentally combined ours with FSW and FSPooling in Task 1, obtaining overall results of 68.45 and 67.81 (ours is 74.86). We preliminarily attribute this gap to the limited expressiveness of these encoders and the weaker alignment (FSPooling) with the SW-based robust objective.
>
> ---
>
> W6:
> Thank you for pointing this out. Our intended claim is our method is effective for the element-level corruption setting we study, `especially under the mild and severe corruption groups across the four tasks`. We will make this claim more precise and less confusing in revision.
>
> ---
>
> Q3:
> Thank you for this insightful question. (1) As reported in Appendix Table 7, Task 1 involves million-scale data, showing that the method is applicable to large-scale SRL. (2) Here we group sets by cardinality |$S_i$​| into four groups: [1, 10], [11, 20], [21, 30], and over. Task 1 results below indicate that the computational overhead grows moderately with set size, remaining practically manageable.
> |R@10(%)/Time(min)|[1,10]|[11,20]|[20,30]| over 30 |
> |---|---|---|---|---|
> |FSW|64.24/11.7| 64.18/11.9| 64.32/12.2 | 64.30/12.7|
> |PSWE|71.44/12.1| 71.81/12.6 | 71.76/13.2 | 71.92/13.9 |
> |Ours|74.65/14.6|74.91/15.9 | 74.77/16.7| 74.81/17.1|
>
> ---
> Q5:
> Thanks for this constructive suggestion. Due to rebuttal space limit, we will provide a complete hyperparameter sensitivity study in the revision.
>
> ---
> *Thank you Reviewer WEba for your review to improve our work. We would be ready to further clarify if any point remains unclear.*

---

### Official Review · Reviewer_iQ9X · 2026-03-17

**Soundness:** 3
**Presentation:** 3
**Significance:** 3
**Originality:** 3
**Overall Recommendation:** 4
**Confidence:** 4

**Summary:**

This paper proposes a distributionally robust optimization framework for Set Representation Learning under inference-time element corruption. The research outlines a core question: how to make set encoders robust when deployed sets contain sparse, label-preserving corruptions absent from training. The authors attempt to address a central concept of intractable worst-case optimization over combinatorial set spaces by modeling sets as empirical measures, defining ambiguity regions via Sliced-Wasserstein distance, and introducing a barycentric adversary that reduces inner maximization to optimization over a low-dimensional simplex. Experiments span four tasks and show consistent improvements over baselines.

**Compliance With Llm Reviewing Policy:**

Affirmed.

**Key Questions For Authors:**

In Proposition 4.1, the neighborhood condition  is enforced at training time. At inference time, the corruption region is defined by SW distance. Are these two \rho values the same? If not, does the theoretical guarantee still hold?

**Limitations:**

Yes

**Strengths And Weaknesses:**

Pros:

1. The problem is well-motivated. Inference-time element corruption is a realistic and underexplored failure mode in SRL.

2. The barycentric adversary is technically clean. Propositions 4.1–4.3 provide a clear theoretical justification and the quantitative gap bound in B.4 is a nice addition.

3. The geometric alignment between SW-based encoding and SW-defined ambiguity region is a principled design choice, validated empirically in Table 5.

Cons:

1. The corruption model assumes label-preserving, sparse element-level degradation. This is a strong assumption. Many real deployments involve structured or correlated corruption (e.g., sensor dropout patterns in point clouds). The method's behavior under non-sparse or label-correlated corruption is not studied.

2. The barycentric adversary operates in embedding space, not input space. The implicit corrupted set is virtual and may not correspond to any realistic input. The authors should discuss whether this limits practical interpretability or adversarial realism.

3. Computational overhead is analyzed theoretically but Table 4 only compares training time on one dataset. Scalability to larger set cardinalities is not validated.

---

> ### Author Rebuttal · Authors · 2026-03-31
>
> Thank you Reviewer iQ9X for the valuable feedbacks. Below we have carefully addressed each point and hope our responses can adequately resolve the concerns.
>
> ---
>
> W1:
>
> Thank you for your insightful summary of our work. We acknowledge that, as you described, our current setting has the boundary that we focus on element-level, sparse, label-preserving corruption, which provides a tractable entry point to study the `robustness research in set representation learning domain`. Regarding the non-sparse or label-correlated corruption scenarios you raise, the primary challenge lies in the difficulty of obtaining such data in practice. Nevertheless, we totally agree, this is apparently an important and well-motivated direction, and we will include a careful discussion of it as future work in the revised version.
>
> ---
>
> W2:
>
> Thank you Reviewer iQ9X for raising this point. We would like to humbly offer the following clarifications:
> - As you said, an adversary operating in embedding space may not correspond to a valid set structure in input space, especially when observed data is scarce.
> - However, our proposed synthesis method is not an arbitrary latent interpolation. As explained in Proposition 4.3, it admits a tighter upper bound on the objective compared to locally discrete cases of using real set elements, providing a principled theoretical justification.
> - More importantly, the role of this adversary is primarily to serve as an adversarial opponent during training; specifically, to provide a tractable and theoretically grounded approximation to the inner maximization of Eq. (7). On the other hand, while generating back to input-level adversarial examples would undoubtedly provide better interpretability, it would also be more difficult and computationally expensive. For this reason, we current consider the embedding space-based adversarial strategy.
>
> We will explicitly discuss these points in the revised version. Thank you for the valuable reminder.
>
> ---
>
> W3:
>
> Thank you for this constructive suggestion. Due to time constraints, we conducted additional experiments on the LiveJ dataset (Task 1), grouping sets by cardinality |$S_i$​| into four groups: [1, 10], [11, 20], [21, 30], and over 30. The results of several competitive models are summarized below:
> |R@10(%)/Time(min)|[1,10]|[11,20]|[20,30]| over 30 |
> |---|---|---|---|---|
> |FSW|64.24/11.7| 64.18/11.9| 64.32/12.2 | 64.30/12.7|
> |PSWE|71.44/12.1| 71.81/12.6 | 71.76/13.2 | 71.92/13.9 |
> |Ours|74.65/14.6|74.91/15.9 | 74.77/16.7| 74.81/17.1|
>
> We notice that our method exhibits a relatively moderate increase in training time across varying set sizes that actually validates its scalability. Given the consistent performance gains, we believe the additional computational overhead is acceptable. We will include more detailed experimental results and analysis in the revised version.
>
> ---
>
>
>
> Q1:
>
> Thank you for raising this important clarification. We would like to respectfully clarify that the symbol \rho appearing near Eq.(6) and Proposition 4.1 is a notation reuse: in both cases, it denotes a distance-based neighborhood intended to ensure that the adversarial training targets remain reasonable within the region. During the actual methodology design process, we proceed progressively with the latter as the primary objective, and the theoretical guarantees therefore still hold. Additionally, we emphasize that at inference time, actually no such adversary needs to be constructed; the adversarial component is only used during training. We will revise the manuscript to more clearly disambiguate this notation and make these points explicit.
>
> ---
>
>
>
> *Thank you Reviewer iQ9X once again for your expertise and support in reviewing our work. Please let us know if there is any part above that remains unclear.*

---

### Official Review · Reviewer_Nm5x · 2026-03-24

**Soundness:** 3
**Presentation:** 3
**Significance:** 3
**Originality:** 3
**Overall Recommendation:** 4
**Confidence:** 3

**Summary:**

This paper proposes a method for improving the robustness of set representation learning. The authors model the perturbation distribution and utilize the worst-case distribution (WD) to search the ambiguity space in order to identify the worst-case perturbation. Their aim is to achieve a Distributionally Robust Set Optimization. They have evaluated their method on four datasets that show that it is robust against couuroption

**Compliance With Llm Reviewing Policy:**

Affirmed.

**Final Justification:**

The quality of the paper is good. However, as I mentioned, the novelty is limited. The proposed methods appear to be extensions of existing approaches rather than fundamentally new contributions.

**Key Questions For Authors:**

Is there any guarantee regarding the claim of this paper that it presents the worst perturbation?

Can you provide an accuracy cost/performance analysis to determine whether this method is truly worthwhile or achieves good results at the expense of high computational costs?

Additionally, could you explain the novelty of this approach and clarify what makes it unique?

**Limitations:**

I believe the authors should provide more analysis and results (discussion) about when the proposed method fails.

Furthermore, the paper needs to explain in more detail the reproducibility aspects that need to be reimplemented.

**Strengths And Weaknesses:**

First, as far as I understood, the proposed method does not guarantee that the worst-case set of perturbations identified is, in fact, the worst. So, maybe the authors overclaimed here.

Second, the formula used to find S', as far as I understand, is computationally expensive. Although attempts have been made to propose solutions for this issue, they do not fully explore the search space to identify the true worst-case scenarios. Therefore, it seems that this should be analyzed further, particularly in the context of ablation studies or experiments,

When we review the results compared to other methods, especially simple pooling, there does not seem to be a significant improvement despite its complexity. This is particularly evident when we examine Table 2.

Regarding novelty, the fundamental idea of identifying the worst-case perturbation is quite similar to working with adversarial examples, although there are some differences that make it less computable. This concept also brings to mind adversarial training. Beyond that, I am curious about how this method performs against various levels of adversarial examples, especially those that are more complex than simple cases.

---

> ### Author Rebuttal · Authors · 2026-03-31
>
> We sincerely thank Reviewer Nm5x for the thoughtful and constructive comments.  Below, we provide point-by-point responses to address the raised concerns.
>
> ---
>
> W1 / Q1:
>
> We appreciate the reviewer for the important comment.
> As the reviewer said, it would be extremely difficult to construct the exact worst-case perturbation from the full ambiguity set.
> Our goal is therefore to `provide a tractable surrogate for this intractable objective via the barycentric adversary`. In the paper, our intention was to describe it as an `approximation` towards such worst-case perturbation, rather than an exact solution. To avoid any possible overclaim, we will revise the wording throughout the paper and explicitly emphasize that it is an approximate version of the worst-case perturbation.
>
> ---
> W2:
>
> Thank you reivewer for raising this important point. As noted, finding the exact $S'$ is computationally intractable because there is no practical exact solution path. One alternative is random combinational sampling (RCS) of elements to explore the search space. Due to time limits, we conducted an experiment on Task 1 LiveJ dataset, using $n$ rounds of sampling in model training as follows.
>
>
> | LiveJ | n=2 | n=5 | n=10|
> |---|---|---|---|
> | RCS | 69.45/21.7min | 70.56/23.5min | 71.13/27.4min |
> | RCS with SW-distance constraint | 69.52/22.6min | 71.81/24.8min | 71.74/28.1min|
>
> The results show that such naive search is however costly while providing limited gains. We will integrate this analysis with Table 4 in Section 5.3.2.
>
> ---
> W3:
>
> We appreciate the reviewer for this critical observation. We `humbly hope to point out that, our goal is to improve robustness under mild/severe inference-time corruption, while preserving clean performance as much as possible`. Although the gains over simple pooling in Table 2 may appear modest in some settings, our method shows stronger advantages on the other three tasks.
> Therefore, while we acknowledge it could be very challenge to simultaneously perform the best across all datasets and task, our proposed method achieves a competitive robustness-accuracy trade-off in the four evaluation scenarios.
>
> ---
> W4:
>
> Thank you for this constructive suggestion. Constructing informative adversarial examples to approximately optimize Eq. (7) constitutes one of the core ideas of our approach.
> Beyond our approach, two broad strategies are possible: (1) gradient-based perturbations and (2) embedding-level perturbations. Due to time constraints, we implemented two representative baselines on Task 1 (LiveJ): FGSM [1] and random Gaussian noise on set embeddings, which achieved overall scores of 72.35 and 74.53, respectively. Our preliminary analysis suggests that these methods are less effective mainly because they are decoupled from the objective in Eq. (7). We will include a more detailed discussion in the revision.
>
> 1. Goodfellow, Ian J., Jonathon Shlens, and Christian Szegedy. "Explaining and harnessing adversarial examples." 2014.
>
> ---
>
> Q2:
>
> Thank you for this question. Compared with competitive baselines on Task 1, the time costs(min) of SetTRSM, FSW, PSWE, Ours are 10.5, 12.3, 13.7, 15.6, respectively. In addition, the costs of different design choices within our framework are reported in Tables 4 and 5. Overall, our method does incur extra computation, but we believe this overhead is acceptable, given the element corruption problem considered in this paper and the competitive performance of our method across multiple tasks. We will make this cost-performance discussion more explicit in the revision.
>
> ---
> Q3:
>
> The novelty of our work is that: (1) we formulate robustness in SRL as a distributionally robust optimization problem over corrupted sets. (2) Since the original objective is intractable, we then progressively derive practical approximations and ultimately introduce a barycentric synthesis mechanism, that enables an efficient approximate solution. Therefore, this formulation and the resulting approximation strategy distinguish our approach from standard adversarial training.
>
> ---
> Limitations:
>
> Thank you for highlighting these issues. First, our method assumes that element corruption is label-preserving at the set level. If the set is multi-labeled or the corruption induces label-level shift, the method may fail; we will discuss it in future work of our revised paper. Second, regarding reproducibility, we have reported implementation details in the appendix, and we will also release the code to further improve reproducibility. Thanks for the reminder.
>
> ---
>
> *We are deeply grateful to Reviewer Nm5x for your time dedicated to improving our work. We have made every effort to address the concerns. Please let us know if there are any remaining points requiring further clarification.*

---

> > ### Author Rebuttal · Reviewer_Nm5x · 2026-04-01
> >
> > I believe this paper is suitable for publication, but I still have some concerns about its novelty. Therefore, I will maintain my positive recommendation but will not be increasing my vote. I understand that novelty can be somewhat subjective, while the value of the paper is clear and measurable. However, I think the method presented is quite similar to previous approaches, as you have essentially built on existing ideas.

---

> > > ### Author Response · Authors · 2026-04-01
> > >
> > > Thank you Reviewer Nm5x for your continued support. We are very glad that our previous responses have addressed all your concerns. If you have any further thoughts you would like to discuss, please feel free to let us know!

---

### Decision · Program_Chairs · 2026-04-30

**Decision:**

Accept (regular)

**Comment:**

This paper proposes a distributionally robust optimization framework for set representation learning under inference-time element corruption. It addresses the challenge that set-based models trained on clean data often fail when a subset of elements is corrupted. The method models sets as empirical measures and defines ambiguity regions using the Sliced-Wasserstein distance. To approximate worst-case perturbations efficiently, the authors introduce a barycentric adversary that reduces the intractable inner maximization to a low-dimensional optimization problem. Experiments on four tasks demonstrate improved robustness to corruption while maintaining performance on clean datasets.

The key contribution of this paper is the proposal of a distributionally robust optimization framework for set representation learning. One of the main concerns was the relationship between real adversarial samples and their surrogate; the authors have addressed this issue well. Overall, the approach is solid and supported by theoretical guarantees, and the reviewers have given positive scores. Therefore, I also recommend this paper for acceptance.